# Free-Rider and Conflict Aware Collaboration Formation for Cross-Silo Federated Learning

**Mengmeng Chen[1], Xiaohu Wu[1], Xiaoli Tang[2], Tiantian He[3,4], Yew-Soon Ong[2,3,4], Qiqi Liu[5], Qicheng Lao[1], Han Yu[2]\***

[1]Beijing University of Posts and Telecommunications, China
[2]College of Computing and Data Science, Nanyang Technological University, Singapore
[3]Institute of High Performance Computing, Agency for Science, Technology and Research, Singapore
[4]Centre for Frontier AI Research, Agency for Science, Technology and Research, Singapore
[5]School of Engineering, Westlake University, China
{chenmengmeng0306,xiaohu.wu,qicheng.lao}@bupt.edu.cn, {xiaoli001,asysong, han.yu}@ntu.edu.sg, he_tiantian@cfar.a-star.edu.sg, liuqiqi@westlake.edu.cn

## Abstract

Federated learning (FL) is a machine learning paradigm that allows multiple FL participants (FL-PTs) to collaborate on training models without sharing private data. Due to data heterogeneity, negative transfer may occur in the FL training process. This necessitates FL-PT selection based on their data complementarity. In cross-silo FL, organizations that engage in business activities are key sources of FL-PTs. The resulting FL ecosystem has two features: (i) self-interest, and (ii) competition among FL-PTs. This requires the desirable FL-PT selection strategy to simultaneously mitigate the problems of free riders and conflicts of interest among competitors. To this end, we propose an optimal FL collaboration formation strategy -FedEgoists- which ensures that: (1) a FL-PT can benefit from FL if and only if it benefits the FL ecosystem, and (2) a FL-PT will not contribute to its competitors or their supporters. It provides an efficient clustering solution to group FL-PTs into coalitions, ensuring that within each coalition, FL-PTs share the same interest. We theoretically prove that the FL-PT coalitions formed are optimal since no coalitions can collaborate together to improve the utility of any of their members. Extensive experiments on widely adopted benchmark datasets demonstrate the effectiveness of FedEgoists compared to nine state-of-the-art baseline methods, and its ability to establish efficient collaborative networks in cross-silos FL with FL-PTs that engage in business activities.

## 1  Introduction

Federated learning (FL) is a promising paradigm of distributed machine learning (ML) as it does not require sharing raw data between FL participants (FL-PTs), thereby upholding the privacy considerations [10, 12, 54, 55, 56]. In the popular Federated Averaging (FedAvg) framework, multiple FL-PTs train a shared model locally with their own datasets, and upload their local model updates to a central server (CS), which then aggregates these model updates and distributes the resulting global model to each FL-PT [32]. There are two types of FL [20]. In *cross-device FL*, FL-PTs are end-user devices such as smartphones or IoT devices, and CS is the final owner of the trained model. In *cross-silo FL*, FL-PTs are companies or organizations in private or public sectors and are the final owners/users of the trained model, while the CS has the authority to coordinate the FL training process.

---

\*Corresponding author

38th Conference on Neural Information Processing Systems (NeurIPS 2024).

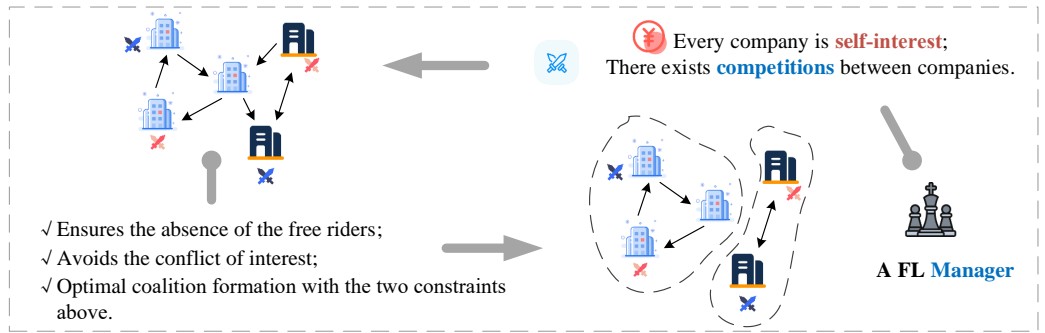

Figure 1: An overview of the main motivation and results of this paper.

We consider in this paper the scenario of cross-silo FL where organizations in the business sector that engage in business activities are key sources of FL-PTs [16]. The business sector is part of the private sector made up by companies and includes business that operate for profit; it is a key driver of technological innovation. The application of FL in the business sector has been studied in diverse domains, including digital banking, ridesharing, recommender systems, and Electric Vehicle charging services [28, 29, 39, 47, 53]. Regional banks have different user groups from their respective regions and are independent [54], while the banks in the same region can compete for users [31]. Recently, a FL platform, MELLODDY, has been developed for drug discovery, including multiple companies from business sectors [35]. Generally, competition exists when there are multiple organizations that are in the same market area and hope to unlock the power of FL [16, 45].

From a FL-PT's perspective, the scenario under study has two features: (i) self-interest and (ii) competition among FL-PTs. In business sectors, individuals are self-interested. This magnifies the free-riding problem where some FL-PTs benefit from the contributions of others without making any contribution to the FL ecosystem [20, 22]. Competition signifies that there is a potential conflict of interest between some two FL-PTs. Thus, two principles can simultaneously be used to meet the individual's needs: (1) a FL-PT can benefit from the FL ecosystem if and only if it can benefit the FL ecosystem, and (2) a FL-PT will not contribute to its competitors as well as the allies of its competitors [42]. The first principle is proposed and formulated in this paper for the first time, while the second principle has previously been considered for other scenarios [42]. From a FL manager's perspective, its priority is to regulate the FL ecosystem to meet the requirements of its customers (i.e., FL-PTs).

This paper studies cross-silo FL in business sectors where there are typically a limited number of FL-PTs (e.g., 2 to 100) [16, 27]. We consider both self-interest and competition for the first time in literature, although it has been well recognized that the free riding problem can be especially serious when self-interested FL-PTs engage in business competition [17, 41]. Since competition and self-interest are considered, all FL-PTs are partitioned into disjoint groups/coalitions, each with the common interest. The competition and benefit relationships among FL-PTs can be described as graphs. We use theoretical tools from graph theory and propose an efficient solution `FedEgoists` that can well satisfy the two principles above. Meanwhile, `FedEgoists` can help FL-PTs achieve the best possible ML model performances, i.e., subject to the two principles, the coalitions that `FedEgoists` finds are optimal in the sense that one coalition cannot increase the utility of any of its members by collaborating with any other coalitions. Extensive experiments over real-world datasets have demonstrated the effectiveness of the proposed solution compared to nine baseline methods, and its ability to establish efficient collaborative networks in cross-silo FL with FL-PTs that engage in business activities. The main motivation and results of this paper are illustrated in Figure 1.

## 2 Related Work

The existing FL works consider the following three lines of works separately and don't address the self-interest and competition features simultaneously.

**Competition.** Collaboration between competing companies is an important research area in business studies [3, 11]. Recently, there has been a growing interest in the study of competition in cross-silo FL where a key application area is the business sector. There are two cases. *In the first case*, all FL-PTs are assumed to offer the same service in the same market area and compete against each other. Several works attempt to reduce the potential side effect after FL-PTs join a fully competitive FL ecosystem. Wu and Yu [50] aim to maintain a negligible change in market share [51, 52] and analyze the achievability of this objective. Tsoy and Konstantinov [45] and Huang et al. [16] study the profitability of FL-PTs, but are taken under different assumptions on the source of extra profit brought by FL. Specifically, Tsoy and Konstantinov [45] assume that: (i) each consumer allocates a fixed budget to multiple services from different markets, and (ii) if a FL-PT owns a better model, the consumer will thus allocate more of its budget to consume its service due to the improved service quality. Huang et al. [16] consider duopoly business competition and assume that, if the model-related service can be improved by FL, customers will have a willingness to pay more and FL-PTs thus have opportunities to increase their profits. *In the second case*, not all FL-PTs compete against each other, like one assumption in this paper, where some FL-PTs come from different market areas. Tan et al. [42] study how to avoid the conflicts of interest among FL-PTs for the first time and give a heuristic to realize Principle 2. However, Tan et al. [42] don't consider the self-interest feature of cross-silo FL in the business sector and cannot guarantee the optimal collaboration among independent coalitions without competition. The collaboration relationships among FL-PTs result in a collaboration graph $\mathcal{G}_u$. In their heuristic, FL-PTs are considered one by one and $\mathcal{G}_u$ is thus constructed gradually to guarantee that for each FL-PT, there is no path that can be reachable to or from its competitors. In this paper, we apply the concepts of cliques and strongly connected components in graph theory to satisfy the self-interest and competition features and further optimize the proposed algorithm.

**Free-riding.** Karimireddy et al. [22] show that a naive scheme can lead to catastrophic levels of free-riding where the benefits of data sharing are completely eroded. By contract theory, they introduce accuracy shaping based mechanisms to prevent free-riding.

**Coalitions.** A coalition is a cooperative relationship formed among different individuals due to certain interests. Each agent is only concerned with the contributions of other agents within the same coalition [1]. Prior studies mainly focus on alleviating the side effect of data heterogeneity by allocating proper collaborators with data complementarity to each FL-PT. Donahue and Kleinberg [9] provide an analytical understanding of what partition of FL-PTs leads to a core-stable coalition structure for mean estimation and linear regression. Cui et al. [5] propose a heuristic algorithm to compute the core-stable coalition structure for general learning tasks. To some extent, the CS can dictate the collaboration relationships among FL-PTs only if doing so can better address the concerns of FL-PTs such as fairness and performance [6, 13, 59, 60]. Chaudhury et al. [4] treat all FL-PTs as a grand coalition and optimize a common model such that there is no other coalition $\mathcal{S}$ of FL-PTs that could significantly benefit more by training a model only using the data of $\mathcal{S}$. Sattler et al. [37] propose a novel federated multi-task learning framework where the geometric properties of the FL loss surface is leveraged to group FL-PTs into clusters with jointly trainable data distributions. Ding and Wang [7] partition a total of $n$ FL-PTs into $K$ groups, where $K \ll n$ and the FL-PTs with similar contributors are assigned to the same group. Then, a common model is returned to each group of FL-PTs such that the performance of using $K$ models for $n$ FL-PTs is a good approximation to the the performance of learning $n$ personalized models for $n$ FL-PTs respectively. However, the above works don't consider the adversarial relationships among FL-PTs.

## 3 Model and Assumptions

### 3.1 Relationship Graphs

There is a set of FL-PTs denoted by $\mathcal{V} = \{v_1, v_2, \cdots, v_n\}$. Each FL-PT $v_i$ is uniquely equipped with its own local dataset $\mathcal{D}_i$. A benefit graph $\mathcal{G}_b$ is deployed to describe the potential collaboration advantages between any two FL-PTs. Together with the competing relationships represented by $\mathcal{G}_c$ among different FL-PTs, we pave the way for the formation of the contribution graph $\mathcal{G}_u$. The benefit graph $\mathcal{G}_b$ is a weighted directed graph with an edge set $E_b$ (i.e., $\mathcal{G}_b = (\mathcal{V}, E_b)$) and is used to evaluate the data complementarity between FL-PTs. It is defined as follows. For any two FL-PTs $v_j$ and $v_i$, if $v_i$ can benefit from $v_j$'s data, then there is a directed edge from $v_j$ to $v_i$ (i.e., $(v_j, v_i) \in E_b$) and the weight of this edge is $w_{j,i} > 0$ where a larger value of $w_{j,i}$ signifies a larger benefit to $v_i$ brought by

$v_j$. In contrast, if $v_i$ cannot benefit from the data of $v_j$, then $(v_j, v_i) \notin E_b$ and $w_{j,i} = 0$. The benefit graph can be computed by the hypernetwork technique like [5, 33, 42]. The competing graph $\mathcal{G}_c$ is an undirected graph with an edge set $E_c$, i.e., $\mathcal{G}_c = (\mathcal{V}, E_c)$. For any two FL-PTs $v_i$ and $v_j$, if they compete against each other, then there is an undirected edge between $v_i$ to $v_j$ (i.e., $(v_i, v_j) \in E_c$). If they are independent of each other, then $(v_i, v_j) \notin E_c$. The CS is assumed to be trustable. In the real world, the CS may represent an impartial and authoritative third-party (e.g., the industry association) [5]. Then, FL-PTs can report their competitive relationships to the third-party in person and confidentiality agreements can be signed between the third-party and FL-PTs. Each FL-PT $v_i$ will report its competitors to CS, as it hopes that CS will correctly utilize this information to prevent its competitors from benefiting from its data. Thus, CS has the knowledge of $\mathcal{G}_c$. Although $v_i$ may benefit from $v_j$'s data($w_{j,i} > 0$), CS has the authority to determine whether $v_i$ can actually utilize $v_j$'s local model update information (i.e., indirectly use $v_j$'s data) in the FL training process or not. Let $\mathcal{X} = (x_{j,i})$ be a $n \times n$ matrix where $x_{j,i} \in \{0, 1\}$: for two different FL-PTs $v_i$ and $v_j$, $x_{j,i}$ is set to one if $v_j$ will contribute to $v_i$ (i.e., $v_i$ will utilize $v_j$'s local model update information) in the FL training process and $x_{j,i}$ is set to zero otherwise. $\mathcal{X}$ defines a directed graph $\mathcal{G}_u = (\mathcal{V}, E_u)$, called the data usage graph: $(v_j, v_i) \in E_u$ if and only if $j \neq i, x_{j,i} = 1$; then, $v_j$ is said to be a collaborator or contributor of $v_i$.

Finally, we introduce some concepts in graph theory. Let $\mathcal{G} = (\mathcal{V}, E)$ denote an arbitrary graph whose node set is $\mathcal{V}$ and whose edge set is $E$. For any subset $\mathcal{S} \subseteq \mathcal{V}$, *an (induced) subgraph $\mathcal{G}(\mathcal{S})$ of the graph $\mathcal{G}$ is such that (i) the node set of $\mathcal{G}(\mathcal{S})$ is $\hat{\mathcal{V}}$ and (ii) the edge set of $\mathcal{G}(\mathcal{S})$ consists of all of the edges in $E$ that have both endpoints in $\mathcal{S}$.* A subgraph $\mathcal{G}(\mathcal{S})$ is said to be a *strongly connected component* of $\mathcal{G}$ if we have (i) $\mathcal{G}(\mathcal{S})$ is strongly connected, i.e., there is a path in each direction between any two nodes and (ii) $\mathcal{G}(\mathcal{S})$ is maximal in the sense that no additional edges or nodes from $\mathcal{G}$ can be included in the subgraph without breaking the property of being strongly connected. The collection of strongly connected components forms a partition of the nodes of $\mathcal{G}$. A simple path is a path in a graph which does not have repeating nodes.

## 3.2 Collaboration Principles

The FL-PTs in the scenario under study have two features: (i) self-interest and (ii) competition among FL-PTs. Accordingly, there are two collaboration principles introduced below.

### 3.2.1 Absence of free riders

While establishing collaboration relationships among FL-PTs, the following principle is used to guarantee the non-existence of free rider.

**Principle 1.** *For any FL-PT $v_i \in \mathcal{V}$, there exists a FL-PT $v_j \in \mathcal{V}$ that benefits $v_i$ if and only if there exists at least one FL-PT $v_k$ that can benefit from $v_i$.*

In this paper, all Fl-PTs are assumed to be self-interested and there exists competition among some FL-PTs. Let $\pi$ denote a partition of FL-PTs $\mathcal{V}$ into $K$ mutually disjoint groups where $\pi = \{\mathcal{S}_1, \mathcal{S}_2, \cdots, \mathcal{S}_K\}$ satisfies: (i) $\mathcal{S}_i \subseteq \mathcal{V}$, (ii) $\cup_{k=1}^{K} \mathcal{S}_k = \mathcal{V}$ and (iii) $\mathcal{S}_i \cap \mathcal{S}_j = \emptyset, \forall i, j \in \{1, 2, ..., K\}$. For any $\mathcal{S}_k \in \pi$, the collaboration relationships among the FL-PTs of $\mathcal{S}_k$ are established according to the subgraph $\mathcal{G}_b(\mathcal{S}_k)$, i.e., for any $v_i, v_j \in \mathcal{S}_k$, $v_j$ is a contributor of $v_i$ if and only if there exists an edge $(v_j, v_i)$ in the graph $\mathcal{G}_b(\mathcal{S}_k)$. Each FL-PT $v_i \in \mathcal{S}_k$ is only concerned with the contributions of other FL-PTs within the same $\mathcal{S}_k$.

**Definition 1** (Coalitions). *A partition $\pi = \{\mathcal{S}_1, \mathcal{S}_2, \cdots, \mathcal{S}_K\}$ is said to be a set of coalitions if we have for any $\mathcal{S}_k \in \pi$ with $|\mathcal{S}_k| \geqslant 2$ and $v_i \in \mathcal{S}_k$ that*

$$\sum_{v_j \in \mathcal{S}_k - \{v_i\}} w_{i,j} > 0 \text{ and } \sum_{v_j \in \mathcal{S}_k - \{v_i\}} w_{j,i} > 0 \tag{1}$$

In a coalition $\mathcal{S}_k$ with $|\mathcal{S}_k| \geqslant 2$, each FL-PT $v_i \in \mathcal{S}_k$ can both benefit and benefit from other members in $\mathcal{S}_k - \{v_i\}$. Thus, in a set $\pi$ of coalitions, Principle 1 will be realized since Eq. (1) is satisfied. If $|\mathcal{S}_k| = 1$, the single FL-PT of $\mathcal{S}_k$ neither benefit nor benefit from any other FL-PT.

### 3.2.2 Avoiding conflict of interest

To avoid conflict of interest, a FL-PT will not contribute to its competitors (i.e., its enemies) and any FL-PTs that help its competitors (i.e., the friends of its enemies). Also, it doesn't hope to see

others help the supporters/friends of its competitors (in)directly. For any FL-PT $v_i$, let $\mathcal{A}_s$ denote the supporter alliance of $v_i$ defined as: $\mathcal{A}_i = \{v_k \in \mathcal{V} \mid v_k \text{ is reachable to } v_i \text{ in the data usage graph } \mathcal{G}_u\}$. Let us consider any two competing FL-PTs $v_j$ and $v_i$ where $(v_j, v_i) \in E_c$. To avoid conflict of interest, $v_j$ will not contribute to any member of $\mathcal{A}_i$. Like [42], this is defined as the following principle by which it is strictly guaranteed that no FL-PTs will contribute to its competitors (in)directly.

**Principle 2.** *For any two competing FL-PTs $v_i$ and $v_j$, $v_j$ is unreachable to $v_i$ in the data usage graph $\mathcal{G}_u$.*

### 3.3 Problem Description

Now, we propose a proper problem formulation such that the resulting solution can well satisfy the FL-PTs' needs and help them achieve the best possible ML model performances. All FL-PTs are divided into a set $\pi$ of coalitions. For the FL-PTs of of a coalition $\mathcal{S}_k \in \pi$, their collaboration relationships are established according to the subgraph $\mathcal{G}_b(\mathcal{S}_k)$. The utility $u(\mathcal{S}_k)$ of a coalition $\mathcal{S}_k$ is defined as the sum of the edge weights of $\mathcal{G}_b(\mathcal{S}_k)$. Formally, let $E_b(\mathcal{S}_k)$ denotes the edge set of $\mathcal{G}_b(\mathcal{S}_k)$; then, $u(\mathcal{S}_k) = \sum_{(v_j, v_i) \in E_b(\mathcal{S}_k)} w_{j,i}$; given a FL-PT $v_i \in \mathcal{S}_k$, its utility is defined as $u_i(\mathcal{S}_k) = \sum_{(v_j, v_i) \in E_b(\mathcal{S}_k)} w_{j,i}$ where $v_i$ is fixed; here, the utility of a coalition is the sum of the utilities of all its members, i.e., $u(\mathcal{S}_k) = \sum_{v_i \in \mathcal{S}_k} u_i(\mathcal{S}_k)$. For two coalitions $\mathcal{S}_k$ and $\mathcal{S}'_k$ with $\mathcal{S}_k \subsetneq \mathcal{S}'_k$, $\mathcal{G}_b(\mathcal{S}_k)$ is a subgraph of $\mathcal{G}_b(\mathcal{S}'_k)$; then the utility of $v_i$ in a larger coalition $\mathcal{S}'_k$ is no smaller than its utility in a smaller coalition $\mathcal{S}_k$, i.e., $u_i(\mathcal{S}_k) \leqslant u_i(\mathcal{S}'_k)$.

In this paper, our final problem is to find a partition $\pi$ of FL-PTs such that

- Principles 1 and 2 are satisfied.
- Subject to Principles 1 and 2, no coalitions of $\pi$ (i.e., no subset $\pi'$ of $\pi$) can collaborate together and be merged into a larger coalition with a higher utility; in other words, after merging these coalitions into a larger one, no FL-PTs in these coalitions $\pi'$ increase their utilities under the larger coalition. Formally, let

$$\Pi = \left\{ \pi' \subseteq \pi \mid \sum_{\mathcal{S}_k \in \pi'} u(\mathcal{S}_k) < u\left( \bigcup_{\mathcal{S}_k \in \pi'} \mathcal{S}_k \right), \text{ Principles 1 and 2 are satisfied by } \bigcup_{\mathcal{S}_k \in \pi'} \mathcal{S}_k \right\}.$$

Then, $\Pi$ satisfies:

$$\Pi = \emptyset. \tag{2}$$

## 4 Solution

Now, we give an algorithm that satisfies Principles 1 and 2 and Eq. (2). It is presented as Algorithm 1 and its main idea is as follows. Firstly, we find a partition $\hat{\pi} = \{\hat{\mathcal{S}}_1, \hat{\mathcal{S}}_2, \cdots, \hat{\mathcal{S}}_H\}$ of all FL-PTs $\mathcal{V}$ such that the FL-PTs of each subset $\hat{\mathcal{S}}_h \in \hat{\pi}$ are independent of each other. Secondly, $\hat{\mathcal{S}}_h \in \hat{\pi}$ is further partitioned into several subsets/coalitions, denoted as $SCC_h = \{\hat{\mathcal{S}}_{h,1}, \hat{\mathcal{S}}_{h,2}, \cdots, \hat{\mathcal{S}}_{h,y_h}\}$ such that for all $l \in [1, y_h]$, $\mathcal{G}_b(\hat{\mathcal{S}}_{h,l})$ is a strongly connected component of $\mathcal{G}_b(\hat{\mathcal{S}}_h)$. Thirdly, for any coalitions of $\bigcup_{h=1}^H SCC_h$, we merge these coalitions into a larger one if doing so achieves a higher coalition utility without violating Principles 1 and 2.

Below, we detail Algorithm 1. Firstly, we define $\mathcal{G}_c^- = (\mathcal{V}, E_c^-)$ as the inverse of the competing graph $\mathcal{G}_c$, i.e., any two nodes of $\mathcal{G}_c^-$ are adjacent if and only if they are not adjacent in $\mathcal{G}_c$; if there is an edge between two nodes in $\mathcal{G}_c^-$, then these two nodes are said to be independent of each other (line 2). In graph theory, a clique of an arbitrary undirected graph $\mathcal{G} = (\mathcal{V}, E)$ is a subset of nodes $\mathcal{S} \subseteq \mathcal{V}$ in which any two nodes of $\mathcal{S}$ are connected by an edge in the subgraph $\mathcal{G}(\mathcal{S})$; a maximal clique is a clique such that if it is extended by adding any other node, the resulting larger subgraph is not complete. We search for all maximal cliques within $\mathcal{G}_c^-$, which form a partition $\hat{\pi}$ of all FL-PTs $\mathcal{V}$. Any two FL-PTs in a subset $\hat{\mathcal{S}}_k \in \hat{\pi}$ are independent of each other. In line 3, the cliques of $\mathcal{G}_c^-$ can be found by the classic Bron–Kerbosch algorithm [44].

Secondly, for each $\hat{\mathcal{S}}_h \in \hat{\pi}$, we process the subgraph $\mathcal{G}_b(\hat{\mathcal{S}}_h)$ of the benefit graph. All strongly connected components of $\mathcal{G}_b(\hat{\mathcal{S}}_h)$ are founded by the classic Tarjan algorithm [43]. The node sets of

**Algorithm 1:** Conflict-free Coalitions without Free Riders

**Input:** The benefit graph $\mathcal{G}_b$, the competing graph $\mathcal{G}_c$
**Output:** The set $\pi$ of coalitions

1   $\pi \leftarrow \emptyset$;         `// Record the set of coalitions found by this algorithm.`

2   Construct the inverse of $\mathcal{G}_c$, denoted as $\mathcal{G}_c^-$;

3   Find all maximal cliques of $\mathcal{G}_c^-$, denoted as $\hat{\pi} = \left\{ \hat{\mathcal{S}}_1, \cdots, \hat{\mathcal{S}}_H \right\}$, by the Bron–Kerbosch algorithm;

4   **for** $h \leftarrow 1$ **to** $H$ **do**

5      Find all strongly connected components of $\mathcal{G}_b\left(\hat{\mathcal{S}}_h\right)$ by the Tarjan algorithm; `// The node`
       `sets of the components of` $\mathcal{G}_b\left(\hat{\mathcal{S}}_h\right)$ `are denoted as` $SCC_h = \left\{ \hat{\mathcal{S}}_{h,1}, \cdots, \hat{\mathcal{S}}_{h,y_h} \right\}$.

6   Let $\pi = \{\hat{v}_1, \hat{v}_2, \cdots, \hat{v}_Y\} = \bigcup_{h=1}^{H} SCC_h$ where $Y = \sum_{h=1}^{H} y_h$;

7   Construct by Definition 2 a directed graph $\mathcal{Z}_b$ and an undirected graph $\mathcal{Z}_c$ whose node sets are
    $\pi$;         `//` $\hat{v}_y$ `is a node in` $\mathcal{Z}_b$ `and` $\mathcal{Z}_c$ `but also represents a subset of` $\mathcal{V}$.
    `/* Below, the node` $\hat{v}_l$ `of` $\mathcal{Z}_b$ `with` $|\hat{v}_l| = 1$ `is processed.`        `*/`

8   Let $y \leftarrow Y + 1$;       `//` $y$ `is the index of the new node` $\hat{v}_y$ `to be constructed.`

9   $(\pi, \mathcal{Z}_b, \mathcal{Z}_c, y) \leftarrow \text{MergeCycle}(\pi, \mathcal{Z}_b, \mathcal{Z}_c, y)$, presented as Algorithm 2;

10   $(\pi, \mathcal{Z}_b, \mathcal{Z}_c, y) \leftarrow \text{MergePath}(\pi, \mathcal{Z}_b, \mathcal{Z}_c, y)$, presented as Algorithm 4;
    `/* Below, the edge` $(\hat{v}_l, \hat{v}_{l'})$ `of` $\mathcal{Z}_b$ `with` $|\hat{v}_l| \geqslant 2$ `and` $|\hat{v}_{l'}| \geqslant 2$ `is processed.`     `*/`

11   $(\pi, \mathcal{Z}_b, \mathcal{Z}_c, y) \leftarrow \text{MergeNeighbors}(\pi, \mathcal{Z}_b, \mathcal{Z}_c, y)$, presented as Algorithm 5;

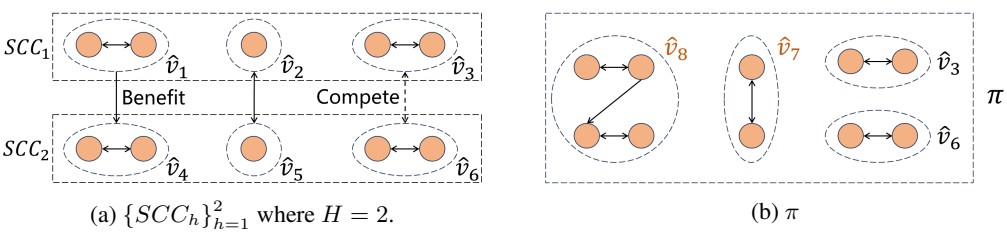

(a) $\{SCC_h\}_{h=1}^2$ where $H = 2$.                          (b) $\pi$

Figure 2: Illustration of Algorithm 1.

all these components constitute a partition of $\hat{\mathcal{S}}_h$, denoted as $SCC_h$ (lines 4-5). Thirdly, we also use $\hat{v}_y$ to denote a set $\hat{\mathcal{S}}_{h,y_h} \subseteq \mathcal{V}$ and let $\pi = \bigcup_{h=1}^{H} SCC_h = \{\hat{v}_1, \hat{v}_2, \cdots, \hat{v}_Y\}$ where $Y = \sum_{h=1}^{H} y_h \leq n$ and $\cup_{y=1}^{Y} \hat{v}_y = \mathcal{V}$ (line 6); then we construct by Definition 2 two new graphs $\mathcal{Z}_b$ and $\mathcal{Z}_c$ whose node sets are $\pi$ (line 7).

**Definition 2.** *In the graph $\mathcal{Z}_b$, there is a directed edge from $\hat{v}_l$ to $\hat{v}_{l'}$ if and only if there exist two nodes $v_i \in \hat{v}_l$ and $v_j \in \hat{v}_{l'}$ such that $(v_i, v_j)$ is a directed edge in the benefit graph $\mathcal{G}_b$. In the graph $\mathcal{Z}_c$, there is an undirected edge between $\hat{v}_l$ and $\hat{v}_{l'}$ if and only if there exist two nodes $v_i \in \hat{v}_l$ and $v_j \in \hat{v}_{l'}$ such that $(v_i, v_j)$ is an undirected edge in the competing graph $\mathcal{G}_c$. For any two coalitions $\hat{v}_l$ and $\hat{v}_{l'}$ of $\pi$, $\hat{v}_l$ is said to benefit (resp. benefit from) $\hat{v}_{l'}$ if there is a directed edge $(\hat{v}_l, \hat{v}_{l'})$ (resp. $(\hat{v}_{l'}, \hat{v}_l)$) in the graph $\mathcal{Z}_b$; $\hat{v}_l$ and $\hat{v}_{l'}$ are said to be competitive if there is an undirected edge $(\hat{v}_l, \hat{v}_{l'})$ in the graph $\mathcal{Z}_c$ and independent of each other otherwise.*

In line 9, Algorithm 2 is called where there is a while-loop. In Algorithm 2, we check whether the loop condition in line 1 is satisfied or not; this loop condition will be called *the cycle condition* subsequently. As illustrated by $\hat{v}_2$ and $\hat{v}_5$ in Figure 2(a), if "yes", let $\mathcal{X} = \{\hat{v}_{y_i}\}_{i=1}^{\theta}$ denote the nodes of this cycle. The following operations are taken to update the graph $\mathcal{Z}_b$ and $\mathcal{Z}_c$: (i) the nodes of $\mathcal{X}$ are merged into a new node $\hat{v}_y$ where all the edges in the graph $\mathcal{Z}_b$ that point to (resp. point from) the nodes of $\mathcal{X}$ change to point to (resp. point from) $\hat{v}_y$, and (ii) the nodes of $\mathcal{X}$ in $\mathcal{Z}_b$ also exist in the graph $\mathcal{Z}_c$; such nodes in $\mathcal{Z}_c$ are still merged into a new node $\hat{v}_y$, and all the edges in the graph $\mathcal{Z}_c$ whose endpoints are the nodes of $\mathcal{X}$ change to become the edges whose endpoints are $\hat{v}_y$. The above operations will be used as a routine and are given in Algorithm 3.

In line 10, Algorithm 4 is called where there are two while-loops. In Algorithm 4, we check whether the loop condition in the outer while loop is satisfied (line 1); this loop condition will be called *the*

**Algorithm 2:** MergeCycle($\pi, \mathcal{Z}_b, \mathcal{Z}_c, y$)

---

**1 while** *there is a node $\hat{v}_{y_i}$ of $\mathcal{Z}_b$ with $|\hat{v}_{y_i}| = 1$ such that (i) there is a cycle $(\hat{v}_{y_1}, \hat{v}_{y_2}, \cdots, \hat{v}_{y_\theta}, \hat{v}_{y_1})$ in the graph $\mathcal{Z}_b$ that contains $\hat{v}_{y_i}$ and (ii) the nodes $\hat{v}_{y_1}, \cdots, \hat{v}_{y_\theta}$ of this cycle are independent of each other* **do**
   // the cycle condition.
**2**   $(\hat{v}_y, \pi, \mathcal{Z}_b, \mathcal{Z}_c, y) \leftarrow \mathrm{Merge}(\mathcal{X}, \pi, \mathcal{Z}_b, \mathcal{Z}_c, y)$, presented as Algorithm 3, where $\mathcal{X} = \{\hat{v}_{y_l}\}_{l=1}^\theta$;
**3 Return** $(\pi, \mathcal{Z}_b, \mathcal{Z}_c, y)$;

---

**Algorithm 3:** Merge($\mathcal{X}, \pi, \mathcal{Z}_b, \mathcal{Z}_c, y$)

---

**1** $\hat{v}_y \leftarrow \bigcup_{\hat{v}_j \in \mathcal{X}} \hat{v}_j$, $y \leftarrow y + 1$, $\pi \leftarrow \pi - \mathcal{X}$, and $\pi \leftarrow \pi \cup \{\hat{v}_y\}$;
**2** Add $\hat{v}_y$ into $\mathcal{Z}_b$ as a new node, and all the edges in the graph $\mathcal{Z}_b$ that point to (resp. point from) the nodes of $\mathcal{X}$ change to point to (resp. point from) $\hat{v}_y$;
**3** Add $\hat{v}_y$ into $\mathcal{Z}_c$ as a new node, and all the edges in the graph $\mathcal{Z}_c$ whose endpoints are the nodes of $\mathcal{X}$ change to become the edges whose endpoints are $\hat{v}_y$;
**4** Remove the nodes of $\mathcal{X}$ from both $\mathcal{Z}_b$ and $\mathcal{Z}_c$;
**5 Return** $(\hat{v}_y, \pi, \mathcal{Z}_b, \mathcal{Z}_c, y)$;

---

**Algorithm 4:** MergePath($\pi, \mathcal{Z}_b, \mathcal{Z}_c, y$)

---

**1 while** *there is a node $\hat{v}_{y_i}$ of $\mathcal{Z}_b$ with $|\hat{v}_{y_i}| = 1$ such that (i) there is a simple path $(\hat{v}_{y_1}, \cdots, \hat{v}_{y_i}, \cdots, \hat{v}_{y_\theta})$ with $\hat{v}_{y_1} \geqslant 2$ and $\hat{v}_{y_\theta} \geqslant 2$ and (ii) the nodes $\hat{v}_{y_1}, \cdots, \hat{v}_{y_\theta}$ of this simple path are independent of each other* **do**
   // the path condition.
**2**   $(\hat{v}_y, \pi, \mathcal{Z}_b, \mathcal{Z}_c, y) \leftarrow \mathrm{Merge}(\mathcal{X}, \pi, \mathcal{Z}_b, \mathcal{Z}_c, y)$ where $\mathcal{X} = \{\hat{v}_{y_l}\}_{l=1}^\theta$;
**3**   $(\pi, \mathcal{Z}_b, \mathcal{Z}_c, y) \leftarrow \mathrm{MergeCycle}(\pi, \mathcal{Z}_b, \mathcal{Z}_c, y)$, presented as Algorithm 2;
**4 Return** $(\pi, \mathcal{Z}_b, \mathcal{Z}_c, y)$;

---

**Algorithm 5:** MergeNeighbors($\pi, \mathcal{Z}_b, \mathcal{Z}_c, y$)

---

**1 while** *there is an edge $(\hat{v}_l, \hat{v}_{l'})$ of $\mathcal{Z}_b$ with $|\hat{v}_l| \geqslant 2$ and $|\hat{v}_{l'}| \geqslant 2$ such that $\hat{v}_l$ and $\hat{v}_{l'}$ are independent of each other* **do**
   // the node condition.
**2**   $(\hat{v}_y, \pi, \mathcal{Z}_b, \mathcal{Z}_c, y) \leftarrow \mathrm{Merge}(\mathcal{X}, \pi, \mathcal{Z}_b, \mathcal{Z}_c, y)$ where $\mathcal{X} = \{\hat{v}_l, \hat{v}_{l'}\}$;
**3**   $(\pi, \mathcal{Z}_b, \mathcal{Z}_c, y) \leftarrow \mathrm{MergeCycle}(\pi, \mathcal{Z}_b, \mathcal{Z}_c, y)$, presented as Algorithm 2;
**4**   $(\pi, \mathcal{Z}_b, \mathcal{Z}_c, y) \leftarrow \mathrm{MergePath}(\pi, \mathcal{Z}_b, \mathcal{Z}_c, y)$, presented as Algorithm 4;
**5 Return** $(\mathcal{Z}_b, \mathcal{Z}_c, y)$;

---

*path condition*. If "yes", let $\mathcal{X} = \{\hat{v}_{y_i}\}_{i=1}^\theta$ denote the nodes of the simple path. Then, Algorithm 3 is called with the current $\mathcal{X}$ as its input. Afterwards, Algorithm 2 is called here to remove that type of cycles defined in line 1 of Algorithm 2. In line 11, Algorithm 5 is called where there are three while-loops. We check whether the loop condition in the outer while loop is satisfied (line 1); this loop condition is called *the node condition*. As illustrated by $\hat{v}_1$ and $\hat{v}_4$ in Figure 2(a), if "yes", let $\mathcal{X} = \{\hat{v}_l\} \cup \{\hat{v}_{l'}\}$; then, Algorithm 3 is called with $\mathcal{X}$ as its input. Afterwards, Algorithms 2 and 4 are called here to guarantee that those types of cycles and simple paths in the cycle and path conditions don't exist in the graph $\mathcal{Z}_b$.

The time complexity of algorithm 1 depends on the Bron-Kerbosch algorithm in line 3 and is $\mathcal{O}(3^{n/3})$ [44]; please see Appendix A.1 for a detailed analysis. In line 1 of Algorithm 2 or 4, any two coalitions $\hat{v}_{y_l}$ and $\hat{v}_{y_{l'}}$ are independent (of each other); in line 1 of Algorithm 5, the two coalitions $\hat{v}_l$ and $\hat{v}_{l'}$ are independent. Any two members from such two coalitions respectively will be independent by Definition 2. In each coalition, all its members are independent. Thus, all members of the merged $\hat{v}_y$ are independent of each other. For any coalition $\hat{v}_y$, the collaboration relationships among its members are established according to the subgraph $\mathcal{G}_b(\hat{v}_y)$; thus, we have two conclusions. Firstly, Principle 2 is realized since for the data usage graph $\mathcal{G}_u$, paths may exist only between two nodes

in a coalition. Secondly, after the merge operation in line 2 of Algorithm 2 or 4, for any $\hat{v}_{y_l}$ with $|\hat{v}_{y_l}| = 1$, it can benefit from its direct predecessor and benefit its direct successor in the cycle or the simple path, satisfying Eq. (1). As a result, we have the following proposition whose detailed proof can be found in Appendix A.2.

**Proposition 1.** *Upon completion of Algorithm 1, Principles 1 and 2 are realized.*

Upon completion of Algorithm 1, the cycle condition, the path condition, and the node condition are not satisfied. Then, we have the following two conclusions: (i) when the cycle condition isn't satisfied, we have for any coalition $\hat{v}_{y_i}$ with $|\hat{v}_{y_i}| = 1$ that it can be merged with other coalitions without violating Principles 1 and 2 if and only if the path condition is satisfied in the graphs $\mathcal{Z}_b$ and $\mathcal{Z}_c$; and (ii) When both the cycle condition and the path condition aren't satisfied, we have for any coalition $\hat{v}_l$ with $|\hat{v}_l| \geqslant 2$ that it can be merged with other coalitions without violating Principles 1 and 2 if and only if the node condition is satisfied in the graphs $\mathcal{Z}_b$ and $\mathcal{Z}_c$. The proofs of the above two conclusions are given in Appendix A.3. With these two conclusions, we further have the following proposition.

**Proposition 2.** *Upon completion of Algorithm 1, Eq. (2) holds.*

## 5 Evaluation

### 5.1 Experimental Setup

**Datasets & data heterogeneity.** We conduct experiments on the CIFAR-10 and CIFAR-100 datasets with different data heterogeneity settings [23]. Also, we use a real-world dataset eICU [36] to illustrate the practicality of `FedEgoists`. Both the **CIFAR-10** and **CIFAR-100** datasets contain 60,000 color images for image classification tasks but have different levels of complexity. CIFAR-10 images have 10 classes with 6,000 images per class, while CIFAR-100 is more complex and has 100 classes with only 600 images per class. We simulate the data heterogeneity by two typical approaches: (i) pathological distribution [5, 32, 42, 61], where each FL-PT is randomly allocated 2 classes of images for CIFAR-10 and 20 classes of images for CIFAR-100, and (ii) Dirichlet distribution [41, 57, 62], where a distribution vector $q_c \in \mathbb{R}^n$ is drawn from the distribution $Dir_n(\beta)$ for each class $c$ and FL-PT $v_i$ is allocated a $q_{c,i}$ proportion of data samples of class $c$; smaller $\beta$ value results in higher data heterogeneity, and we set $\beta = 0.5$. Above, we use standard datasets and randomly divide each dataset into different parts that are used as the local data of different FL-PTs. We also use a real-world dataset that contains data directly from multiple different FL-PTs. Specifically, **eICU** is a dataset collecting EHRs from many hospitals across the United States admitted to the intensive care unit (ICU). The task is to predict mortality during hospitalization. More details on the way of processing these datasets are given in Appendix B.1.

**Comparison baselines.** **FedAvg** [32] is widely recognized as a vanilla FL algorithm. Multiple representative personalized FL (PFL) methods are used as baselines for comparison [41]. **FedProx** [25] and **SCAFFOLD** [21] represent two typical approaches that make the aggregated model at the CS close to the global optima and are two benchmarks in [24]. In other baseline methods, different personalized models are trained for individual FL-PTs. **pFedHN** [38] and **pFedMe** [40] represent two approaches based on hypernetworks and meta-learning respectively. **FedDisco** [58] and **pFedGraph** [57] are two approaches based on data complementarity, which are proposed recently. **FedOra**[48] is a method that assesses whether a FL-PT's generalization performance can benefit from knowledge transferred from others and maximizes it. The last baseline is **Local** where each FL-PT simply takes local ML training without collaboration. These traditional FL approaches cannot address the issues of both self-interest and competition. Thus, we use the operations in lines 1-4 of Algorithm 1 to generate a set of coalitions, denoted as $\cup_{h=1}^{H} SCC_h$ where $SCC_h = \{\hat{\mathcal{S}}_{h,1}, \cdots, \hat{\mathcal{S}}_{h,y_h}\}$. If $|\hat{\mathcal{S}}_{h,1}| \geqslant 2$, $\hat{\mathcal{S}}_{h,1}$ represents a group of independent FL-PTs that can collaborate together following Principles 1 and 2. the above nine FL approaches are applied to each group of such FL-PTs to generate the baseline results.

Finally, like [5, 26, 42], the hypernetwork technique in [33] is used to compute the benefit graph $\mathcal{G}_b$ and a hypernetwork is constructed by a multilayer perceptron (MLP). The specific way of generating $\mathcal{G}_b$ is introduced in Appendix B.2. The used network structures are introduced in Appendix B.3. There are $n = 10$ FL-PTs in the CIFAR-10, CIFAR-100 and eICU experiments.

Table 1: Accuracy comparisons (MTA) under different $\alpha$ on CIFAR10.

| $\alpha$ | | LOCAL | FEDAVG | FEDPROX | SCAFFOLD | PFEDME | PFEDHN | FEDDISCO | PFEDGRAPH | FEDORA | FEDEGOISTS |
|---|---|---|---|---|---|---|---|---|---|---|---|
| **0.05** | PAT | 80.47±2.06 | 36.86±3.00 | 36.62±6.17 | 36.61±6.18 | 48.66±6.38 | 66.53±2.00 | 36.61±6.18 | 52.04±8.66 | 69.73±1.62 | **81.35±0.30** |
| **0.05** | Dir | 61.59±0.53 | 44.98±1.91 | 46.94±2.12 | 46.76±2.92 | 44.64±2.61 | 55.61±0.45 | 46.74±2.99 | 46.56±2.55 | 55.28±0.75 | **63.06±0.64** |
| **0.1** | PAT | 80.47±2.06 | 49.40±5.50 | 48.19±5.17 | 48.18±5.16 | 56.56±1.66 | 66.61±1.62 | 48.19±5.17 | 55.35±4.51 | 68.65±2.02 | **80.73±1.35** |
| **0.1** | Dir | 61.59±0.53 | 46.77±1.96 | 48.71±1.97 | 48.61±2.02 | 46.65±2.74 | 54.21±0.83 | 48.56±1.99 | 49.10±3.19 | 55.97±0.22 | **62.74±1.09** |
| **0.2** | PAT | 80.47±2.06 | 63.67±2.10 | 57.26±1.48 | 57.24±2.34 | 79.27±1.35 | 76.08±2.20 | 57.25±2.15 | 60.27±2.33 | 72.74±1.91 | **81.30±1.46** |
| **0.2** | Dir | 61.59±0.53 | 55.69±1.90 | 53.79±1.07 | 54.16±0.79 | 53.64±0.79 | 61.31±0.56 | 54.08±1.43 | 53.85±1.07 | 55.67±0.96 | **66.62±1.23** |
| **0.3** | PAT | 80.47±2.06 | 57.95±2.37 | 59.82±4.88 | 59.83±4.87 | 63.09±3.26 | 65.11±2.4 | 59.82±4.88 | 62.12±4.51 | 71.51±2.40 | **81.37±1.41** |
| **0.3** | Dir | 61.59±0.53 | 50.48±0.87 | 49.99±1.15 | 50.09±1.29 | 49.33±1.94 | 53.21±0.49 | 50.17±1.29 | 50.66±1.59 | 55.9±1.01 | **63.39±0.89** |
| **0.4** | PAT | 80.47±2.06 | 58.47±5.87 | 63.28±4.54 | 63.27±4.54 | 66.36±3.88 | 67.51±3.04 | 63.28±4.55 | 63.30±4.61 | 72.89±1.67 | **82.54±0.30** |
| **0.4** | Dir | 61.59±0.53 | 50.14±2.2 | 51.20±2.16 | 51.23±2.09 | 51.00±0.94 | 53.04±0.80 | 51.14±2.09 | 51.14±2.16 | 57.26±0.32 | **62.81±0.88** |

Table 2: Accuracy comparisons(MTA) under different $\alpha$ on CIFAR100.

| $\alpha$ | | LOCAL | FEDAVG | FEDPROX | SCAFFOLD | PFEDME | PFEDHN | FEDDISCO | PFEDGRAPH | FEDORA | FEDEGOISTS |
|---|---|---|---|---|---|---|---|---|---|---|---|
| **0.05** | PAT | 46.24±1.38 | 34.52±8.65 | 35.42±1.36 | 35.47±1.36 | 35.78±1.72 | 29.98±1.07 | 35.42±3.58 | 36.60±1.15 | 41.91±0.49 | **47.00±1.81** |
| **0.05** | Dir | **30.31±0.48** | 15.33±5.35 | 19.81±6.54 | 19.73±6.50 | 18.71±1.41 | 18.12±0.92 | 19.76±6.56 | 19.76±6.50 | 27.06±0.26 | 27.59±1.52 |
| **0.1** | PAT | 46.24±1.38 | 40.01±0.89 | 42.57±0.44 | 42.73±0.44 | 34.40±4.67 | 30.17±0.47 | 42.56±0.45 | 42.78±0.46 | 42.63±1.04 | **46.28±1.05** |
| **0.1** | Dir | 30.31±0.48 | 20.25±4.93 | 18.86±5.07 | 18.80±5.03 | 20.51±0.98 | 17.45±0.55 | 18.87±5.05 | 18.88±4.95 | 27.50±0.21 | **32.01±1.66** |
| **0.2** | PAT | 46.24±1.38 | 29.68±4.12 | 28.60±4.56 | 28.55±4.34 | 29.90±1.85 | 28.38±0.71 | 29.05±4.11 | 30.51±4.03 | 41.63±1.65 | **50.21±2.24** |
| **0.2** | Dir | 30.31±0.48 | 19.24±1.13 | 20.10±0.35 | 20.00±0.48 | 19.89±0.36 | 23.11±0.79 | 19.93±0.38 | 20.17±0.35 | 27.24±0.36 | **32.86±1.53** |
| **0.3** | PAT | 46.24±1.38 | 40.24±0.55 | 42.42±0.42 | 42.57±0.30 | 44.34±2.16 | 29.63±0.23 | 42.42±0.41 | 42.48±0.48 | 41.72±1.98 | **46.38±1.83** |
| **0.3** | Dir | 30.31±0.48 | 25.56±0.32 | 27.37±0.17 | 27.27±0.24 | 25.28±2.55 | 17.21±0.17 | 27.37±0.17 | 26.18±1.69 | 27.43±0.20 | **34.30±0.44** |
| **0.4** | PAT | 46.24±1.38 | 40.52±0.27 | 41.63±1.03 | 41.71±1.05 | 44.38±1.94 | 30.18±0.28 | 41.73±1.03 | 41.66±1.10 | 42.94±0.25 | **48.16±1.61** |
| **0.4** | Dir | 30.31±0.48 | 24.73±0.97 | 27.37±0.40 | 27.31±0.26 | 26.72±1.89 | 17.08±0.35 | 27.37±0.40 | 27.17±0.42 | 27.24±0.23 | **34.15±0.96** |

## 5.2 Benchmark Experiments: CIFAR-10 & CIFAR-100

We conduct experiments on CIFAR-10 and CIFAR-100 with competing graphs that are generated randomly. To simulate competition, the probability of two FL-PTs competing against each other is set to $\alpha$, thus generating a random competing graph $\mathcal{G}_c$ [42], which constrains the collaboration between some FL-PTs; here, the probability that two FL-PTs are independent of each other is 1-$\alpha$. The value of $\alpha$ determines the intensity of competition among FL-PTs and a larger value reflects a higher level of competing intensity among FL-PTs.

Firstly, we verify the effect of the competition intensity on the effectiveness of the proposed solution where we vary the value of $\alpha$ that takes different values in $\{0.05, 0.1, 0.2, 0.3, 0.4\}$ and conducted the corresponding experiments. Given a specific value of $\alpha$, we conducted five trials to show the average performance. For the $l$-th trial, a particular competing graph $\mathcal{G}_{c,l}$ is randomly generated with the given $\alpha$; then, the experiments for the baseline and proposed approaches are run; the performance of the proposed approach is denoted as $r_{\alpha,l,p}$ while the performance of the $l$-th baseline approach is denoted as $r_{\alpha,l,i}$, $i \in \{1, 2, \cdots, 9\}$. Given the value of $\alpha$, we show the average performance of the five trials (i.e., $\sum_{l=1}^{5} \frac{r_{\alpha,l,p}}{5}$ and $\sum_{l=1}^{5} \frac{r_{\alpha,l,i}}{5}$, $i \in \{1, 2, \cdots, 9\}$). The model performance is measured by the mean test accuracy (MTA), and the experimental results are presented in Tables 1–2 in the form of mean±std; here, *PAT* and *Dir* represent the pathological and Dirichlet distributions respectively. It is observed from Tables 1–2 that on average, FedEgoists achieves a significant performance improvement when compared with all the baseline approaches. In Appendix B.4.1, we also illustrate the collaboration relationships among the ten FL-PTs. It is shown that `FedEgoists` can facilitate the collaboration among FL-PTs and thus achieves a better performance.

Secondly, we define a metric to evaluate the worst-case performance across the five trials. In each trial, the nine baseline approaches perform differently. We find an integer $l^*$ such that under the $l^*$-th trial, the baseline approaches achieve the best performance, i.e. $l^* = \arg\max_{l \in [1,5]} \left( \max_{i \in [1,9]} r_{\alpha,l,i} \right)$ where $\max_{i \in [1,9]} r_{\alpha,l,i}$ is the best performance of all the nine baseline approaches in the $l$-th trial. In the trial that is the most advantageous to the baseline approaches, $\max_{i \in [1,9]} r_{\alpha,l^*,i} - r_{\alpha,l^*,p}$ is the performance improvement (or difference) of the best baseline approach to the proposed approach, which may be negative if the proposed approach achieves a better performance. Upon identifying $l^*$, we then detail the performance outcomes of both the proposed and baseline models during the $l^*$-th trial. For various values of $\alpha$, ranging from 0.05 to 0.4, we calculate the performance difference, $\max_{i \in [1,9]} r_{\alpha,l^*,i} - r_{\alpha,l^*,p}$. The results for these computations, under each specified $\alpha$,

Table 3: The worst-case performance of the proposed approach compared with the baseline approaches.

| | 0.05 | | 0.1 | | 0.2 | | 0.3 | | 0.4 | |
|---|---|---|---|---|---|---|---|---|---|---|
| | PAT | Dir | PAT | Dir | PAT | Dir | PAT | Dir | PAT | Dir |
| CIFAR10 | 0.011000 | -0.002903 | 0.022900 | -0.000624 | 0.025800 | -0.0006030 | 0.028800 | -0.005725 | -0.002399 | -0.000100 |
| CIFAR100 | -0.000999 | 0.076002 | 0.011400 | -0.000008 | -0.000636 | -0.0009356 | -0.000020 | -0.032153 | -0.000699 | -0.027078 |

Table 4: eICU

| AUC | LOCAL | FEDAVG | FEDPROX | SCAFFOLD | pFEDME | pFEDHN | FEDDISCO | pFEDGRAPH | FEDORA | FedEgoists |
|---|---|---|---|---|---|---|---|---|---|---|
| $v_0$ | 53.64±22.12 | 63.52±22.40 | 80.42±9.85 | 80.24±9.92 | 52.30±19.79 | 41.94±19.14 | 60.48±13.07 | 80.42±9.85 | **90.36±2.26** | 66.36±19.28 |
| $v_1$ | 67.94±6.88 | 62.55±16.49 | 57.03±16.62 | 57.21±16.68 | 46.00±34.96 | 76.61±14.77 | 63.76±14.97 | 59.62±7.49 | 81.52±16.91 | **81.58±6.65** |
| $v_2$ | 37.33±17.74 | 76.48±12.70 | 60.13±6.77 | 60.38±6.64 | 36.48±27.59 | 79.62±16.18 | **92.70±4.60** | 57.32±8.17 | 47.56±9.62 | 66.04±33.21 |
| $v_3$ | 79.88±21.16 | 67.04±26.74 | 78.74±15.66 | 78.87±15.44 | 45.79±32.04 | 55.35±26.55 | 80.38±18.24 | 78.69±7.48 | 75.12±7.85 | **84.40±5.76** |
| $v_4$ | 52.48±11.61 | 73.46±15.58 | 73.63±9.74 | 75.75±11.07 | 57.07±23.12 | 48.75±22.68 | 70.15±9.96 | 49.61±5.31 | 48.95±6.80 | **75.84±11.26** |
| $v_5$ | 39.45±9.06 | 57.09±7.46 | 61.94±9.13 | 61.70±9.12 | 55.15±24.92 | 52.55±25.12 | 53.03±9.73 | **89.37±7.71** | 77.72±8.24 | 68.41±5.60 |
| $v_6$ | 68.00±32.62 | 77.61±5.87 | 79.62±7.62 | 78.74±7.81 | 57.23±32.51 | 42.01±16.65 | 82.26±6.41 | **98.80±0.76** | 98.55±1.18 | 56.86±7.52 |
| $v_7$ | 73.36±7.08 | 71.80±9.52 | 73.55±10.48 | 73.59±10.17 | 56.60±7.56 | 51.21±5.01 | 68.45±10.98 | 76.82±11.07 | 75.53±5.94 | **77.97±14.94** |
| $v_8$ | 36.24±22.56 | 73.55±2.70 | 77.47±3.80 | 77.43±3.66 | 61.22±10.49 | 46.71±16.08 | 65.05±3.41 | 69.16±3.12 | 72.26±12.01 | **90.60±10.57** |
| $v_9$ | 71.70±10.64 | 63.14±9.42 | 63.82±9.32 | 63.79±9.36 | 42.97±12.63 | 45.42±17.42 | 63.24±10.63 | 60.76±10.12 | 58.55±7.62 | **79.88±8.29** |
| Avg | 58.01 | 68.62 | 70.66 | 70.77 | 51.08 | 54.02 | 69.95 | 72.06 | 72.61 | **74.79** |

are systematically presented in Table 3, which shows that in the worst case, the proposed FedEgoists has a performance very close to the best performance of all the baseline approaches.

Finally, we also conduct experiments to show the effect of data heterogeneity and the related results can be found in Appendix B.4.2.

## 5.3 Real-world Collaboration Example: eICU

Following the setting in [5, 42], there are ten hospitals in total, with $\{v_i\}_{i=0}^{4}$ as large hospitals and $\{v_i\}_{i=5}^{9}$ as small hospitals. Due to the extreme imbalance of data labels, where over 90% are negative labels, we use the AUC scores to evaluate the performance of the trained model. Suppose there are more than one large hospital located in the same city while small hospitals are dispersed in different rural areas with lower population densities; competition mainly occurs among large hospitals. We posit that competitive relationships exist between the following pairs of FL-PTs: $(v_1, v_4)$, $(v_2, v_3)$, and $(v_2, v_4)$. Table 4 presents our experimental results. In Appendix B.5, we also illustrate the collaboration relationships among the 10 FL-PTs. In the naive baseline methods, the formed coalitions are $\{v_0, v_3, v_4\}$, $\{v_1, v_2\}$ and $\{v_5, v_6, \cdots, v_9\}$. In the proposed method, larger coalitions are formed so that their members can benefit from more FL-PTs in the same coalition; the formed coalitions are $\{v_0, v_3, v_4, \cdots, v_9\}$ and $\{v_1, v_2\}$. Thus, FedEgoists can facilitate the collaboration among FL-PTs. In Table 4, it is observed that FedEgoists achieves a better performance on the whole.

In addition to CIFAR-10, CIFAR-100 and eICU, we also conduct experiments on the synthetic data like [42] and the related settings and results are presented in Appendix B.6.

## 6 Conclusions

In this paper, we consider cross-silos FL where organizations in the business sector that engage in business activities are key sources of FL-PTs. The resulting FL ecosystem has two features: (i) self-interest and (ii) competition among FL-PTs. Correspondingly, two principles are proposed to build a sustainable FL ecosystem: (1) a FL-PT can benefit from the FL ecosystem if and only if it can benefit the FL ecosystem, and (2) a FL-PT will not contribute to its competitors as well as the supporters of its competitors. The Fl ecosystem needs to be tailored to realize these principles. In this paper, we propose an efficient solution that can well realize the two principles above. In the meantime, for any two coalitions without competition that can collaborate together, one coalition cannot increase its utility by collaborating with the other coalition. Extensive experiments over real-world datasets have demonstrated the effectiveness of the proposed solution compared to nine baseline methods, and its ability to establish efficient collaborative networks in cross-silos FL with FL-PTs that engage in business activities.

## Acknowledgments and Disclosure of Funding

This work is supported in part by the National Key R&D Program of China under Grant 2024YFE0200500. This research is supported in part by the National Natural Science Foundation of China under 62327801. This research is supported, in part, by the National Research Foundation Singapore and DSO National Laboratories under the AI Singapore Programme (No: AISG2-RP-2020-019); and the RIE 2020 Advanced Manufacturing and Engineering (AME) Programmatic Fund (No. A20G8b0102), Singapore. This research/project is supported by the National Research Foundation, Singapore and Infocomm Media Development Authority under its Trust Tech Funding Initiative. Any opinions, findings and conclusions or recommendations expressed in this material are those of the author(s) and do not reflect the views of National Research Foundation, Singapore and Infocomm Media Development Authority.

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

## A  Theoretical Analysis

### A.1  The Time Complexity of Algorithm 1

Now, we analyze the time complexity of Algorithm 1. Firstly, the construction of $\mathcal{G}_c$ in line 2 involves two nested for loops, resulting in a complexity of $\mathcal{O}(n^2)$. In line 3, the Bron-Kerbosch algorithm is used, which has a time complexity of $\mathcal{O}(3^{n/3})$ [44]. In lines 4-5, there is a for-loop; each iteration, the Tarjan's algorithm is executed on $\mathcal{G}_b(\hat{\mathcal{S}}_h)$ with a time complexity of $\mathcal{O}(m_h + e_h)$ [43], where $m_h$ is the number of nodes in $\mathcal{G}_b(\hat{\mathcal{S}}_h)$ and $e_h$ denotes the number of its edges; thus, the aggregate time complexity is $\mathcal{O}(n + |E_b|)$. The time complexity of forming a union of sets $\pi$ in line 6 is $\mathcal{O}(Y)$. The construction of $\mathcal{Z}_b$ and $\mathcal{Z}_c$ in line 7 involves two nested loops to check the existence of edges between different node sets; at each iteration, the time complexity is $\mathcal{O}(|\hat{v}_y||\hat{v}_{y'}|)$ where $y, y' \in [1, Y]$; the total time complexity in line 7 is $\mathcal{O}(Y^2 n^2)$ where $|\hat{v}_y| \leqslant n$.

Initially, there are $Y$ elements in the set $\pi = \{\hat{v}_1, \hat{v}_2, \cdots, \hat{v}_Y\}$ where $Y \leqslant n$. $\pi$ is the node sets of $\mathcal{Z}_b$ and $\mathcal{Z}_c$. In lines 9, 10 and 11 of Algorithm 1, Algorithms 2, 4 and 5 are sequentially called. In each of Algorithms 2, 4 and 5, there is at least one while loop and the loop conditions are about the features of a cycle, a simple path or two neighboring nodes in the graph $\mathcal{Z}_b$. At each iteration of these while loops, at least two nodes will be merged into a new node $\hat{v}_y$ with multiple nodes of $\pi$ removed. So, there are a total of at most $Y - 1$ iterations in all these while loops and we always have $|\pi| \leqslant n$; afterwards, Algorithm 1 ends.

In Algorithms 2, 4 and 5, Algorithm 3 is called. The time complexity of line 1 in Algorithm 3 is $\mathcal{O}(|\mathcal{X}|)$. In lines 2-4, $\mathcal{Z}_b$ and $\mathcal{Z}_c$ are reconstructed and the complexity is $\mathcal{O}(d|\mathcal{X}|)$, where $d$ is the maximum of the in-degrees and out-degrees of all nodes of $\mathcal{X}$ in the graphs $\mathcal{Z}_b$ and $\mathcal{Z}_c$ and is upperly bounded by $|\pi|$, which is no larger than $n$. To sum up, the total complexity of Algorithm 3 is $\mathcal{O}(n|\mathcal{X}|) \leqslant \mathcal{O}(n^2)$.

In Algorithm 2, there is a while loop. In line 1, all cycles can be found by the Johnson's algorithm [19], which is implemented in a Python library and has a time complexity of $\mathcal{O}\left((|\pi| + E(\mathcal{Z}_b))(c + 1)\right) \leqslant \mathcal{O}\left((n + E(\mathcal{Z}_b))(c + 1)\right)$ where $E(\mathcal{Z}_b)$ denotes the number of edges in the graph $\mathcal{Z}_b$ and $c$ denotes an upper bound of the total number of cycles discovered. For each cycle, we check whether any two of its nodes are competitive, which has a time complexity $\mathcal{O}(|\pi|^2) \leqslant \mathcal{O}(n^2)$; in the meantime, we check whether there is a node $\hat{v}_{y_i}$ in this cycle with $|\hat{v}_{y_i}| = 1$, which has a time complexity $\mathcal{O}(|\pi|) \leqslant \mathcal{O}(n)$. So, the time complexity of checking the cycle condition in line 1 is $\mathcal{O}\left((n + E(\mathcal{Z}_b))(c + 1)\right) + \mathcal{O}(cn^2) = \mathcal{O}\left((n^2 + E(\mathcal{Z}_b))(c + 1)\right)$. At each iteration of the while loop (line 2), Algorithm 3 is called and the time complexity of Algorithm 3 is $\mathcal{O}(n^2)$. The number of iterations is upperly bounded by $n$. Thus, the complexity of Algorithm 2 is no larger than $\mathcal{O}\left(n(n^2 + E(\mathcal{Z}_b))(c + 1)\right) = \mathcal{O}(f_1(n, E(\mathcal{Z}_b), c))$ where

$$f_1(n, E(\mathcal{Z}_b), c) = n(n^2 + E(\mathcal{Z}_b))(c + 1). \tag{3}$$

In Algorithm 4, there is an outer while loop. In line 1, for any two nodes $\hat{v}_{y_1}$ and $\hat{v}_{y_\theta}$, we check whether they are such that $|\hat{v}_{y_1}| \geqslant 2$ and $|\hat{v}_{y_\theta}| \geqslant 2$, which has time complexity of $\mathcal{O}(|\pi|^2) \leqslant \mathcal{O}(n^2)$. For each pair of $\hat{v}_{y_1}$ and $\hat{v}_{y_\theta}$ with $|\hat{v}_{y_1}| \geqslant 2$ and $|\hat{v}_{y_\theta}| \geqslant 2$, we can use the modified DFS algorithm to find all the simple paths from $\hat{v}_{y_1}$ to $\hat{v}_{y_\theta}$ [8]. This algorithm is implemented in a Python library and has a time complexity of $\mathcal{O}((|\pi| + E(\mathcal{Z}_b))\tau) \leqslant \mathcal{O}((n + E(\mathcal{Z}_b))\tau)$ where $\tau$ denotes an upper bound of the total number of simple paths between two nodes. For each path, we check whether any two of its nodes are competitive, which has a time complexity $\mathcal{O}(|\pi|^2) \leqslant \mathcal{O}(n^2)$; in the meantime, we check whether there is a node $\hat{v}_{y_i}$ in this path with $|\hat{v}_{y_i}| = 1$, which has a time complexity $\mathcal{O}(|\pi|) \leqslant \mathcal{O}(n)$. So, the time complexity of checking the path condition in line 1 is

$\mathcal{O}((n + E(\mathcal{Z}_b))\tau) + \mathcal{O}(\tau n^2) = \mathcal{O}\left((n^2 + E(\mathcal{Z}_b))\tau\right) = \mathcal{O}(f_2(n, E(\mathcal{Z}_b), \tau))$ where

$$f_2(n, E(\mathcal{Z}_b), \tau) = (n^2 + E(\mathcal{Z}_b))\tau. \tag{4}$$

At each iteration of the while loop, the operations in lines 2-3 are executed. In line 2, Algorithm 3 is called and the time complexity of Algorithm 3 is $\mathcal{O}(n^2)$. In line 3, Algorithm 2 is called with a time complexity $\mathcal{O}(f_1(n, E(\mathcal{Z}_b), c))$. The number of iterations is upperly bounded by $n$. Thus, the complexity of Algorithm 4 is $\mathcal{O}(n(f_1(n, E(\mathcal{Z}_b), c) + f_2(n, E(\mathcal{Z}_b), \tau)))$.

In Algorithm 5, the time complexity of checking the node condition in line 1 has a time complexity of $\mathcal{O}(E(\mathcal{Z}_b))$. At each iteration, the operations in lines 2-4 are executed. In lines 2-4, Algorithms 3, 2 and 4 are called sequentially. As analyzed above, these operations have a time complexity no larger than $\mathcal{O}(n(f_1(n, E(\mathcal{Z}_b), c) + f_2(n, E(\mathcal{Z}_b), \tau)))$. The number of iterations is upperly bounded by $n$. Thus, the complexity of Algorithm 5 is no larger than $\mathcal{O}(n^2(f_1(n, E(\mathcal{Z}_b), c) + f_2(n, E(\mathcal{Z}_b), \tau)))$.

To sum up, since a complete graph of $n$ nodes has a total of $n(n-1)$ that is an upper bound of $E(\mathcal{Z}_b)$, where $\pi$ is the node set of $\mathcal{Z}_b$ and $|\pi| \leqslant n$. Further, we have

$$f_1(n, E(\mathcal{Z}_b), c) \leqslant n^3(c + 1) \tag{5}$$

$$f_2(n, E(\mathcal{Z}_b), \tau) = n^3\tau. \tag{6}$$

The worst-case time complexity of the operations in lines 9-11 depends on Algorithm 5 and is no larger than

$$\mathcal{O}\left(n^2(f_1(n, E(\mathcal{Z}_b), c) + f_2(n, E(\mathcal{Z}_b), \tau))\right) \leq \mathcal{O}\left(n^5(\tau + c + 1)\right). \tag{7}$$

Finally, if $c$ and $\tau$ can be upperly bounded by constants, then the time complexity of Algorithm 1 depends on the Bron-Kerbosch algorithm in line 3 and is $\mathcal{O}(3^{n/3})$. We note that this paper studies cross-silo FL in business sectors where there are typically a limited number of FL-PTs (e.g., 2 to 100) [16, 27, 49].

## A.2 Proof of Proposition 1

A graph $\mathcal{G}$ is said to be strongly connected if it contains a directed path from $v_i$ to $v_j$ and a directed path from $v_j$ to $v_i$ for every pair of nodes $v_i$ and $v_j$ of $\mathcal{G}$. Specially, a graph that contains only a single node is said to be strongly connected trivially.

**Lemma 1.** *Principle 1 is realized when Eq. (1) is satisfied.*

*Proof.* Only the FL-PTs in the same coalition may collaborate. For any FL-PT $v_i \in \mathcal{V}$, there exists a coalition $\mathcal{S}_k \in \pi$ such that $v_i \in \mathcal{S}_k$. When $|\mathcal{S}_k| = 1$, the only FL-PT $v_i$ neither receives nor provides benefits to any other FL-PT, thereby satisfying Principle 1 trivially. When $|\mathcal{S}_k| \geqslant 2$, $v_i$ engages in a reciprocal exchange of benefits with the other members of $\mathcal{S}_k - \{v_i\}$, which is guaranteed by Eq. (1). Thus, $v_i$ both contributes to and benefits from other members and Principle 1 is satisfied. ∎

**Lemma 2.** *Upon completion of lines 1-6 of Algorithm 1, we have for any $\hat{v}_l \in \pi$ with $|\hat{v}_l| \geqslant 2$ that (i) all FL-PTs of the coalition $\hat{v}_l$ are independent of each other and (ii) Eq. (1) holds for any FL-PT $v_i \in \hat{v}_l$.*

*Proof.* At line 3, we obtain all maximal cliques of the inverse of $\mathcal{G}_c$, denoted as $\hat{\pi} = \left\{\hat{\mathcal{S}}_1, \cdots, \hat{\mathcal{S}}_H\right\}$. The nodes of $\hat{\mathcal{S}}_h$ are independent of each other. At line 5, all the strongly connected components of $\mathcal{G}_b\left(\hat{\mathcal{S}}_h\right)$ are computed, and the node sets of the components of $\mathcal{G}_b\left(\hat{\mathcal{S}}_h\right)$ are denoted as $SCC_h = \left\{\hat{\mathcal{S}}_{h,1}, \cdots, \hat{\mathcal{S}}_{h,y_h}\right\}$ where $\hat{\mathcal{S}}_h = \bigcup_{l=1}^{y_h} \hat{\mathcal{S}}_{h,l}$. Thus, the FL-PTs of each $\hat{\mathcal{S}}_{h,l}$ are still independent of each other. In line 6, $\pi = \{\hat{v}_1, \hat{v}_2, \cdots, \hat{v}_Y\} = \bigcup_{h=1}^{H} SCC_h$ where $Y = \sum_{h=1}^{H} y_h$. Thus, the first point of Lemma 2 holds. The subgraph $\mathcal{G}_b(\hat{v}_l)$ is strongly connected. If $|\hat{v}_l| \geqslant 2$, we have for any node $v_i \in \hat{v}_l$ that there are nodes of $\hat{v}_l$ that can be reachable to and from $v_i$. Thus, the second point of Lemma 2 holds. ∎

In lines 7-11, some coalitions of $\{\hat{v}_1, \hat{v}_2, \cdots, \hat{v}_Y\}$ may be merged as a larger coalition by Algorithm 3, which occurs in lines 9, 10 and 11 of Algorithm 1 where Algorithms 2, 4 and 5 are called,

respectively; Algorithm 2 is also called in Algorithm 4; Algorithms 2 and 4 are called sequentially in Algorithm 5. The merge operation occurs (i.e., Algorithm 3 is executed) if the cycle condition, the path condition and the node condition are satisfied in Algorithms 2, 4 and 5 respectively. Upon each completion of Algorithm 3, $\pi$ is updated: some coalitions will be merged into a new coalition $\hat{v}_y$ that is further added to $\pi$, and then be removed from $\pi$. Like Lemma 2, we still have the following conclusion.

**Lemma 3.** *At the beginning of each execution of Algorithm 3, suppose we have for any $\hat{v}_l \in \pi$ with $|\hat{v}_l| \geqslant 2$ that (i) all FL-PTs of the coalition $\hat{v}_l$ are independent of each other and (ii) Eq. (1) holds for any FL-PT $v_i \in \hat{v}_l$. Then, upon each completion of Algorithm 3, we still have for any $\hat{v}_l \in \pi$ with $|\hat{v}_l| \geqslant 2$ that (i) all FL-PTs of the coalition $\hat{v}_l$ are independent of each other and (ii) Eq. (1) holds for any FL-PT $v_i \in \hat{v}_l$.*

*Proof.* Algorithm 3 is called if the cycle condition, the path condition and the node condition are satisfied in Algorithms 2, 4 and 5 respectively.

In line 1 of Algorithms 2 or 4, any two coalitions $\hat{v}_{y_l}$ and $\hat{v}_{y_{l'}}$ are independent of each other; by Definition 2, any two members from $\hat{v}_{y_l}$ and $\hat{v}_{y_{l'}}$ respectively are independent of each other. In the meantime, for any $\hat{v}_{y_l} \in \pi$, any two members of $\hat{v}_{y_l}$ are independent of each other. Thus, upon completion of the merge operation in line 2 of Algorithms 2 or 4, $\hat{v}_{y_1}, \hat{v}_{y_2}, \cdots, \hat{v}_{y_\theta}$ are merged into a new coalition $\hat{v}_y$, and all FL-PTs of $\hat{v}_y$ are independent of each other. This completes the proof of the first point of Lemma 3 when it comes to Algorithms 2 and 4.

Similarly, in line 1 of Algorithm 5, any two members from $\hat{v}_l$ and $\hat{v}_{l'}$ respectively are independent of each other. In the meantime, for any $\hat{v}_{y_l} \in \pi$, any two members of $\hat{v}_{y_l}$ are independent of each other. Thus, upon completion of the merge operation in line 2 of Algorithm 5, $\hat{v}_l$ and $\hat{v}_{l'}$ are merged into a new coalition $\hat{v}_y$, and all FL-PTs of $\hat{v}_y$ are independent of each other. This completes the proof of the first point of Lemma 3 when it comes to Algorithm 5.

The collaboration relationships among the members of $\hat{v}_y$ are established exactly according to the subgraph $\mathcal{G}_b(\hat{v}_y)$. In line 1 of Algorithms 2 or 4, for any $l \in [1, \theta]$, if $|\hat{v}_{y_l}| \geqslant 2$, the members of $\hat{v}_{y_l}$ already satisfy Eq. (1). Let us consider any two nodes $\hat{v}_l$ and $\hat{v}_{l'}$ in the graph $\mathcal{Z}_b$. By Definition 2, if there is an edge from $\hat{v}_l$ to $\hat{v}_{l'}$ in the graph $\mathcal{Z}_b$, then there is a node of $\hat{v}_{l'}$ that benefits from a node of $\hat{v}_l$ in the benefit graph $\mathcal{G}_b$. If $|\hat{v}_{y_l}| = 1$, the single node in $\hat{v}_{y_l}$ will benefit its direct successor in the cycle (resp. the simple path) and benefit from its direct predecessor in this cycle (resp. the simple path); thus the node of $\hat{v}_{y_l}$ also satisfies Eq. (1). This completes the proof of the second point of Lemma 3 when it comes to Algorithms 2 and 4. In line 1 of Algorithm 5, since $|\hat{v}_l| \geqslant 2$ and $|\hat{v}_{l'}| \geqslant 2$, the members of $\hat{v}_l$ and $\hat{v}_{l'}$ already satisfy Eq. (1). This completes the proof of the second point of Lemma 3 when it comes to Algorithm 5. ∎

Finally, by Lemma 2, we have the following conclusion holds upon completion of lines 1-6 of Algorithm 1: for any $\hat{v}_l \in \pi$ with $|\hat{v}_l| \geqslant 2$, (i) all FL-PTs of the coalition $\hat{v}_l$ are independent of each other and (ii) Eq. (1) holds for any FL-PT $v_i \in \hat{v}_l$. Afterwards, Algorithm 3 may be executed multiple times by the call by Algorithms 2, 4 and 5; upon each completion of Algorithm 3, $\pi$ is updated such that some of its coalitions are removed from $\pi$ and merged into a larger one that is added to $\pi$. By Lemma 3, upon completion of Algorithm 1, the conclusion above still holds. Since Eq. (1) holds, Principle 1 is realized by Lemma 1. For any $\hat{v}_l \in \pi$, the collaboration relationships among the members of a coalition $\hat{v}_l$ are established exactly according to the subgraph $\mathcal{G}_b(\hat{v}_l)$. Each FL-PT $v_i$ of a coalition only collaborate with other FL-PTs within the same coalition, The data usage graph $\mathcal{G}_u$ in fact consists of multiple separated subgraphs $\mathcal{G}_b(\hat{v}_l)$, i.e., $\mathcal{G}_u = \{\mathcal{G}_b(\hat{v}_l) \mid \hat{v}_l \in \pi\}$. In the graph $\mathcal{G}_u$, paths may exist only between two FL-PTs of the same coalition, and any two FL-PTs in the same coalition are independent of each other. Thus, Principle 2 is realized.

## A.3 Proof of Proposition 2

After finishing the operations in line 9 of Algorithm 1, the loop condition in line 1 of Algorithm 2 (i.e., the cycle condition) is not satisfied. The execution in line 10 of Algorithm 1 starts. Before executing the first iteration of the while loop of Algorithm 4, if the loop condition (i.e., the path condition) is not satisfied, then Algorithm 4 ends with both the cycle condition and the path condition violated. If Algorithm 4 ends after executing several iterations, then the cycle condition (line 3) and the path condition (line 1) are both violated.

Subject to Principles 1 and 2, no coalitions of $\pi$ (i.e., no subset $\pi'$ of $\pi$) can collaborate together and be merged into a larger coalition with a higher utility; in other words, after merging these coalitions into a larger one, no FL-PTs in these coalitions $\pi'$ increase their utilities under the larger coalition. Formally, like what we define in Section 3.3, let

$$
\Pi = \left\{ \pi' \subseteq \pi \mid \sum_{\hat{v}_{y_i} \in \pi'} u(\hat{v}_{y_i}) < u \left( \bigcup_{\hat{v}_{y_i} \in \pi'} \hat{v}_{y_i} \right), \text{ Principles 1 and 2 are satisfied by } \bigcup_{\hat{v}_{y_i} \in \pi'} \hat{v}_{y_i} \right\}.
$$

Given a graph $\mathcal{Z}_b$ where the cycle and path conditions are not satisfied, we have the following conclusion.

**Lemma 4.** *Suppose the graph $\mathcal{Z}_b$ is a graph without any such cycle that (i) the nodes of this cycle are all independent of each other and (ii) there exists a node $\hat{v}_k$ in this cycle with $|\hat{v}_k| = 1$. Let us consider an arbitrary node $\hat{v}_{y_i}$ of $\mathcal{Z}_b$ with $|\hat{v}_{y_i}| = 1$; each node $\hat{v}_{y_i}$ also represents a coalition; then, we have*

- *The coalition $\hat{v}_{y_i}$ can benefit or benefit from other coalitions by being merged with them without violating Principles 1 and 2 (i.e., there exists a non-empty $\pi' \in \Pi$ such that $\hat{v}_{y_i} \in \pi'$) if and only if there exists a path $(\hat{v}_{y_1}, \cdots, \hat{v}_{y_i}, \cdots, \hat{v}_{y_\theta})$ in the graph $\mathcal{Z}_b$ with $\hat{v}_{y_1} \geqslant 2$ and $\hat{v}_{y_\theta} \geqslant 2$ and the nodes of this path are independent of each other.*

*Proof.* Firstly, we prove the "only if" direction by contradiction. For a coalition $\hat{v}_{y_i}$ that contains only one FL-PT/node, it can collaborate with other coalitions without violating Principle 1 only if there are two other coalitions $\hat{v}_{y_{i-1}}$ and $\hat{v}_{y_{i+1}}$ that can both benefit and benefit from it; by Definition 2, in the case that $\hat{v}_{y_{i-1}}$ and $\hat{v}_{y_{i+1}}$ denote the same coalition, there is a cycle in the graph $\mathcal{Z}_b$ between $\hat{v}_{y_i}$ and $\hat{v}_{y_{i-1}}$ where $\hat{v}_{y_{i-1}} = \hat{v}_{y_{i+1}}$. Suppose there are some coalitions that can collaborate with $\hat{v}_{y_i}$ without violating Principles 1 and 2 (i.e., there exists a non-empty $\pi' \in \Pi$ such that $\hat{v}_{y_i} \in \pi'$); correspondingly, in the graph $\mathcal{Z}_b$, there will be a path of the form $(\hat{v}_{y_1}, \cdots, \hat{v}_{y_{i-1}}, \hat{v}_{y_i}, \hat{v}_{y_{i+1}}, \cdots, \hat{v}_{y_\theta})$ in which a coalition will benefit its successor; the nodes of this path are also independent of each other according to the graph $\mathcal{Z}_c$ in order to be able to collaborate together without violating Principle 2. Then, this path is simple and not a cycle (i.e., $\hat{v}_{y_1} \neq \hat{v}_{y_\theta}$); otherwise, it is a cycle that has two nodes that compete against each other. In this path, all intermediate nodes can both benefit and benefit from other nodes. Each of $\hat{v}_{y_1}$ and $\hat{v}_{y_\theta}$ also represents a coalition and should contain at least two nodes of $\mathcal{V}$; then, each node in $\hat{v}_{y_1}$ and $\hat{v}_{y_\theta}$ already satisfies Eq. (1), thus maintaining Principle 1. Otherwise, if $|\hat{v}_{y_1}| = 1$, then the single node of $\hat{v}_{y_1}$ cannot benefit from others; if $|\hat{v}_{y_\theta}| = 1$, then the single node of $\hat{v}_{y_\theta}$ cannot benefit others. These two cases violate Principle 1. This completes the proof of the "only if" direction.

Secondly, we prove the "if" direction. If there exists a path $(\hat{v}_{y_1}, \cdots, \hat{v}_{y_i}, \cdots, \hat{v}_{y_\theta})$ in the graph $\mathcal{Z}_b$ with $|\hat{v}_{y_1}| \geqslant 2$ and $|\hat{v}_{y_\theta}| \geqslant 2$ and the nodes of this path are independent of each other, then the coalitions that these nodes in the paths represent can be merged into a new subset $\hat{v}_y$ of nodes of $\mathcal{V}$; doing so doesn't violate Principle 2, i.e., the edge set of $\mathcal{G}_c(\hat{v}_y)$ is empty. As shown above, each node in $\hat{v}_{y_1}$ and $\hat{v}_{y_\theta}$ already satisfies Eq. (1) since any two nodes of $\mathcal{G}_b(\hat{v}_1)$ or $\mathcal{G}_b(\hat{v}_\theta)$ are connected. Also, in the graph $\mathcal{Z}_b$, any intermediate node can both benefit and benefit from other nodes in this path without violating Principle 1. Finally, let $\pi' = \{\hat{v}_{y_1}, \cdots, \hat{v}_{y_i}, \cdots, \hat{v}_{y_\theta}\}$. For any $l \in [2, \theta]$, there exists at least one FL-PT $v_i$ of $\hat{v}_{y_l}$ that can benefit from some FL-PT in $\hat{v}_{y_l}$, which is denoted as $v_j$ by Definition 2. Before merging the coalitions of $\pi'$, the FL-PT $v_i$ of a coalition $\hat{v}_{y_l} \in \pi'$ only collaborates with the other FL-PTs with the same coalition $\hat{v}_{y_l}$ and the collaboration relationship is established according to $\mathcal{G}_b(\hat{v}_{y_l})$. Let $\hat{v}_y = \bigcup_{\hat{v}_{y_i} \in \pi'} \hat{v}_{y_i}$. After merging the coalitions of $\pi'$ into a larger coalition $\hat{v}_y$, FL-PT $v_i$ can additionally benefit from FL-PT $v_j$, i.e., $u_i(\hat{v}_{y_l}) < u_i(\hat{v}_y)$; in the subgraph $\mathcal{G}_b(\hat{v}_y)$, $v_i$ can benefit from all FL-PTs of $\hat{v}_y$ such that they point to $v_i$ by directed edges. This completes the proof of the "if" direction. ∎

After finishing the operations in line 10 of Algorithm 1, the cycle condition and the path condition cannot be satisfied. Then, the execution in line 11 of Algorithm 1 starts. Before executing the first iteration of the while loop of Algorithm 5, if the loop condition (i.e., the node condition) is not satisfied, then Algorithm 5 ends with the cycle, path and node conditions violated. If Algorithm 5 ends after executing several iterations, then the cycle and path conditions (lines 3-4) and the node condition (line 1) are violated.

Subject to Principles 1 and 2, no coalitions of $\pi$ (i.e., no subset $\pi'$ of $\pi$) can collaborate together and be merged into a larger coalition with a higher utility; in other words, after merging these coalitions into a larger one, no FL-PTs in these coalitions $\pi'$ increase their utilities under the larger coalition. Formally, like what we define in Section 3.3, let

$$\Pi = \left\{ \pi' \subseteq \pi \mid \sum_{\hat{v}_l \in \pi'} u(\hat{v}_l) < u\left( \bigcup_{\hat{v}_l \in \pi'} \hat{v}_l \right), \text{ Principles 1 and 2 are satisfied by } \bigcup_{\hat{v}_l \in \pi'} \hat{v}_l \right\}.$$

Given a graph $\mathcal{Z}_b$ where the cycle, path and node conditions are not satisfied, we have the following conclusion.

**Lemma 5.** *Suppose the graph $\mathcal{Z}_b$ is a graph without any such path that (i) the nodes of this path are all independent of each other and (ii) this path is either a cycle in which there exists a node $\hat{v}_y$ with $|\hat{v}_y| = 1$, or a simple path of the form $(\hat{v}_{y_1}, \cdots, \hat{v}_{y_i}, \cdots, \hat{v}_{y_\theta})$ with $|\hat{v}_{y_i}| = 1$, $|\hat{v}_{y_1}| \geqslant 2$ and $|\hat{v}_{y_\theta}| \geqslant 2$. Let us consider an arbitrary node $\hat{v}_l$ of $\mathcal{Z}_b$ with $|\hat{v}_l| \geqslant 2$; each node $\hat{v}_l$ also represents a coalition; then, we have*

- *The coalition $\hat{v}_l$ can benefit or benefit from other coalitions by being merged with them without violating Principles 1 and 2 (i.e., there exists a non-empty $\pi' \in \Pi$ such that $\hat{v}_l \in \pi'$) if and only if there exists another node $\hat{v}_{l'}$ with $|\hat{v}_{l'}| \geqslant 2$ that connects $\hat{v}_l$ by an edge in the graph $\mathcal{Z}_b$ and the two nodes $\hat{v}_l$ and $\hat{v}_{l'}$ are independent of each other.*

*Proof.* Firstly, we prove the "if" direction. If there exists another node $\hat{v}_{l'}$ with $|\hat{v}_{l'}| \geqslant 2$ that connects $\hat{v}_l$ by an edge in the graph $\mathcal{Z}_b$ and the two nodes $\hat{v}_l$ and $\hat{v}_{l'}$ are independent of each other, then the coalitions that these two nodes in $\mathcal{Z}_b$ represent can be merged into a new subset $\hat{v}_y$ of nodes of $\mathcal{V}$; doing so doesn't violate Principle 2 by Definition 2. Each FL-PT in $\hat{v}_l$ and $\hat{v}_{l'}$ already satisfies Eq. (1) since any two FL-PTs of $\mathcal{G}_b(\hat{v}_l)$ or $\mathcal{G}_b(\hat{v}_{l'})$ are connected. Thus, any node in the merged set $\hat{v}_y$ still satisfies Principle 1. Finally we have $\pi' = \{\hat{v}_l, \hat{v}_{l'}\}$. This completes the proof of the "if" direction.

Secondly, we prove the "only if" direction. For a coalition $\hat{v}_{y_i}$ that contains only one FL-PT/node, it can collaborate with other coalitions without violating Principle 1 only if there are two other coalitions $\hat{v}_{y_{i-1}}$ and $\hat{v}_{y_{i+1}}$ that can both benefit and benefit from it. Suppose there are some coalitions that collaborate with $\hat{v}_{y_i}$ without violating Principles 1 and 2 (i.e., there exists a non-empty $\pi' \in \Pi$ such that $\hat{v}_l \in \pi'$); correspondingly, in the graph $\mathcal{Z}_b$, there will be a path of the form $(\hat{v}_{y_1}, \hat{v}_{y_2}, \cdots, \hat{v}_{y_i})$, $(\hat{v}_{y_i}, \hat{v}_{y_{i+1}}, \cdots, \hat{v}_{y_\theta})$, or $(\hat{v}_{y_1}, \cdots, \hat{v}_{y_{i-1}}, \hat{v}_{y_i}, \hat{v}_{y_{i+1}}, \cdots, \hat{v}_{y_\theta})$ in which a coalition will benefit its successor; the nodes of this path are also independent of each other according to the graph $\mathcal{Z}_c$ in order to be able to collaborate together without violating Principle 2. Then, by the assumptions in Lemma 5, this path is a simple path in which there doesn't a node such that (i) the coalition that it represents contains only one node of $\mathcal{V}$ and (ii) there don't exist one of its upstream nodes and one of its downstream nodes whose corresponding coalitions both contain at least two nodes $\mathcal{V}$. This means that the coalitions that the nodes of this path represent all contain at least nodes. This completes the proof of the "only if" direction. ■

As shown above, upon completion of Algorithm 1, the cycle, path and node conditions are not satisfied in the graph $\mathcal{Z}_b$. Firstly, by Lemma 4, for arbitrary multiple coalitions in which there is at least one coalition that contains only one FL-PT, they cannot be merged into a larger coalition without violating Principles 1 and 2. Secondly, by Lemma 5, for arbitrary multiple coalitions each of which has at least two FL-PTs, they cannot be merged into a larger coalition without violating Principles 1 and 2. Thus, Proposition 2 holds.

## B  More Experimental Details

### B.1  Data Processing

**CIFAR-10 & CIFAR-100.**   In both CIFAR-10 and CIFAR-100, there are a total of 50,000 images for training and 10,000 images for testing. In CIFAR-10, each class has 5000 training images and 1000 testing images. In CIFAR-100, each class has 500 training images and 100 testing images. In both the CIFAR-10 and CIFAR-100 experiments, we split the data into 10 FL-PTs, and the training data for each FL-PT is divided into a training set (90%) and a validation set (10%). The label distribution of CIFAR-10 is illustrated in Figure 3. In Figure 3, the label distribution across different FL-PTs/FL-PTs indicates that there is a significant variance in the total number of samples per FL-PT, and the data is imbalanced. Similarly, the label distribution of CIFAR-100 is illustrated in Figure 4.

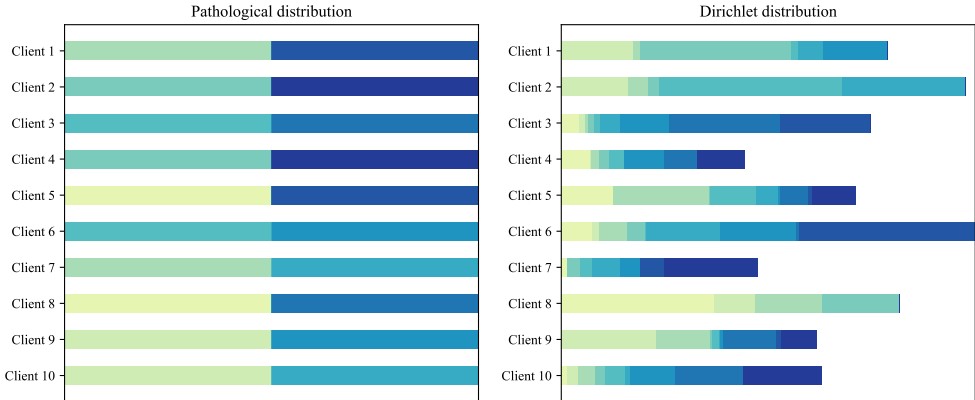

Figure 3: Label distribution of CIFAR-10. Colors indicate the labels for each FL-PT/FL-PT.

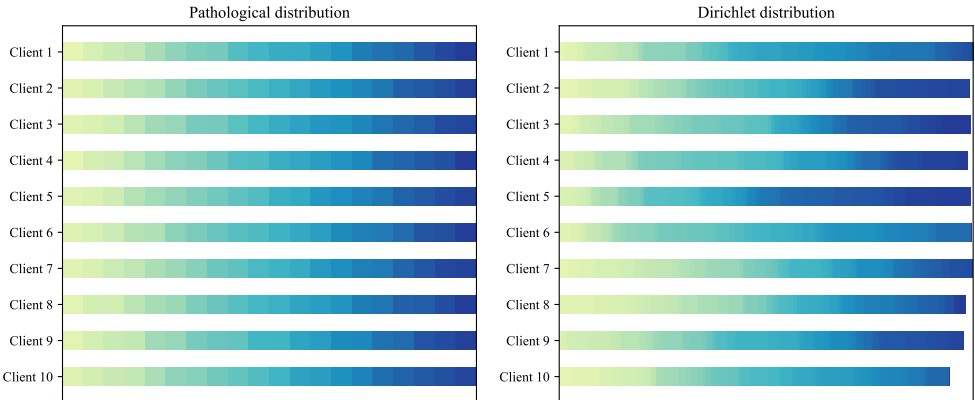

Figure 4: Label distribution of CIFAR-100. Colors indicate the labels for each FL-PT/FL-PT.

**eICU.** We process the eICU dataset following the data pre-processing steps in [18, 30], finally generating JSON files that include hospital identifiers. We divided the training, validation, and testing sets in a ratio of 7:1.5:1.5, with each hospital being treated as an independent FL-PT. We chose a binary classification task using the eICU database, aiming to predict patient mortality within 48 hours of ICU admission($mort\_48h$) based on the initial data provided.

### B.2 The Hypernetwork Technique for Generating the Benefit Graph $\mathcal{G}_b$

The local model information is mainly leveraged in the calculation of $\mathcal{G}_b$. Specifically, there are $n$ FL-PTs. Each FL-PT $v_i$ has a risk/loss function $\ell_i \colon \mathbb{R}^n \to \mathbb{R}_+$. Given a learned hypothesis $h \in \mathcal{H}$, let the loss vector $\ell(h) = [\ell_1, \ldots, \ell_n]$ represent the utility loss of the $n$ FL-PTs under the hypothesis $h$. The hypothesis $h$ is considered a Pareto solution if there is no other hypothesis $h'$ that dominates $h$, i.e.,

$$\nexists h' \in \mathcal{H}, s.\,t.\,\forall i : \ell_i(h') \leqslant \ell_i(h) \text{ and } \exists j : \ell_j(h') < \ell_j(h). \tag{8}$$

Let $r = (r_1, \ldots, r_n) \in \mathbb{R}^n$ denote a preference vector which denotes the weight of the objective local model loss that is normalized with $\sum_{k=1}^{n} r_k = 1$ and $r_k \geq 0, \forall k \in \{1, \ldots, n\}$. The hypernetwork $HN$ takes $r$ as input and outputs a Pareto solution $h$, i.e.,

$$h \leftarrow HN(\phi, r), \tag{9}$$

where $\phi$ denotes the parameters of the hypernetwork [23]. For each FL-PT $v_i$, linear scalarization can be used. Like [5,30], an optimal preference vector $r_i^* = \left(r_{i,1}^*, r_{i,2}^*, \ldots, r_{i,n}^*\right)$ is determined to generate the hypothesis $h_i^*$ that minimizes the loss with the data $\hat{\mathcal{D}}_i$. This is expressed as

$$h_i^* = HN(\phi, r_i^*) \text{ where } r_i^* = argmin_r \hat{\mathcal{L}}_i(HN(\phi, r)). \tag{10}$$

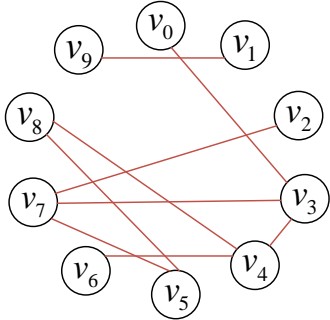

(a) The Competing Graph $\mathcal{G}_c$.

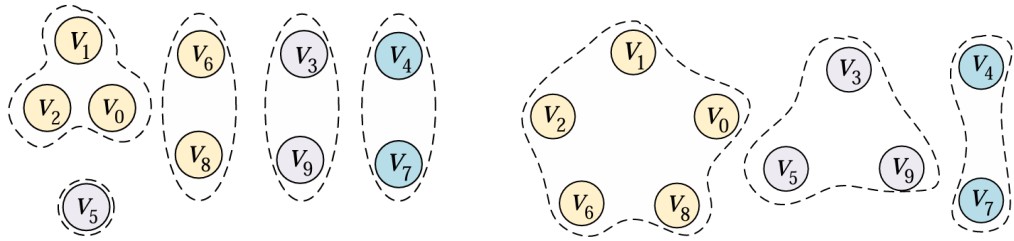

(b) The set $\pi$ of coalitions in the baseline algorithms.

(c) The set $\pi$ of coalitions by Algorithm 1.

Figure 5: **Illustration of Coalitions under CIFAR-100**

For each FL-PT $i$, the value of $r^*_{i,j}$ is used as an estimate to the weight of $v_j$ to $v_i$ [5, 26]. $\{r^*_i\}^n_{i=1}$ defines a directed weighted graph, i.e., the benefit graph $\mathcal{G}_b$.

## B.3 Architecture Details

The network architectures used in different datasets are introduced below [5, 14, 15, 30, 34, 42].

### B.3.1 CIFAR-10 & CIFAR-100

When it comes to a specific dataset, all approaches have the same network structure for each FL-PT to execute the learning tasks.

**Hypernetwork.** We construct a hypernetwork that employs a 2-layer hidden MLP to generate the parameters of the target network.

**CIFAR-10.** The target network passes through two convolutional layers, each followed by a $2 \times 2$ max pooling layer. Afterward, the data is flattened and further processed through two fully connected layers, each followed by a LeakyReLU activation function, finally yielding the prediction results.

**CIFAR-100.** Initially, the input data passes through two convolutional layers, activated by ReLU and each followed by a 2x2 max pooling layer to reduce spatial dimensions and highlight important features. After these convolutional and pooling layers, the data is flattened to prepare for dense processing. Subsequently, it is fed into two fully connected layers, each incorporating a ReLU activation function to add non-linearity and improve learning capabilities. The output is finally produced by a linear layer.

### B.3.2 eICU

We employ a single-layer MLP hypernetwork, while the target network utilizes a Transformer classifier [46] with Layer Normalization [2].

Table 5: Accuracy comparisons under different $\beta$ of Dirichlet distribution

| | CIFAR-10 | | | CIFAR-100 | | |
|---|---|---|---|---|---|---|
| $\beta$ | 0.01 | 0.1 | 0.5 | 0.01 | 0.1 | 0.5 |
| LOCAL | $86.75 \pm 0.10$ | $\mathbf{78.58 \pm 0.80}$ | $58.14 \pm 1.73$ | $\mathbf{58.89 \pm 1.30}$ | $47.23 \pm 0.48$ | $30.92 \pm 0.25$ |
| FEDAVE | $85.22 \pm 0.74$ | $69.68 \pm 3.24$ | $52.25 \pm 1.10$ | $50.13 \pm 1.74$ | $40.05 \pm 2.11$ | $19.85 \pm 1.33$ |
| FEDPROX | $85.85 \pm 0.97$ | $70.17 \pm 1.50$ | $50.35 \pm 1.29$ | $55.25 \pm 1.69$ | $42.38 \pm 2.04$ | $20.71 \pm 0.57$ |
| SCAFFOLD | $85.48 \pm 0.79$ | $70.11 \pm 1.50$ | $50.72 \pm 0.99$ | $55.37 \pm 1.89$ | $42.40 \pm 2.12$ | $20.61 \pm 0.60$ |
| PFEDME | $86.07 \pm 7.34$ | $70.90 \pm 1.69$ | $50.20 \pm 0.90$ | $53.67 \pm 4.07$ | $45.72 \pm 0.89$ | $20.50 \pm 0.58$ |
| PFEDHN | $85.57 \pm 1.65$ | $76.95 \pm 0.56$ | $57.87 \pm 0.78$ | $47.60 \pm 0.48$ | $33.73 \pm 0.31$ | $23.72 \pm 0.91$ |
| FEDDISCO | $85.52 \pm 0.82$ | $70.21 \pm 1.60$ | $50.64 \pm 1.65$ | $55.25 \pm 1.68$ | $42.38 \pm 2.02$ | $20.54 \pm 0.50$ |
| PFEDGRAPH | $85.86 \pm 0.97$ | $70.60 \pm 1.82$ | $50.41 \pm 1.29$ | $55.24 \pm 1.86$ | $42.56 \pm 2.00$ | $20.78 \pm 0.57$ |
| FEDORA | $86.66 \pm 0.07$ | $75.22 \pm 2.07$ | $55.67 \pm 0.96$ | $55.70 \pm 0.61$ | $43.72 \pm 0.79$ | $26.78 \pm 1.06$ |
| FEDEGOISTS | $\mathbf{86.91 \pm 0.07}$ | $78.03 \pm 1.54$ | $\mathbf{63.18 \pm 1.45}$ | $58.06 \pm 1.19$ | $\mathbf{47.72 \pm 3.90}$ | $\mathbf{33.47 \pm 1.75}$ |

## B.4 More Experiments on CIFAR-10 & CIFAR-100

### B.4.1 Illustrating the Collaboration Relationships in the Benchmark Experiments

Now, we give a representative example to illustrate the collaboration relationships among the 10 FL-PTs and present more details on the experimental results. The experiments are taken with CIFAR-100 under the Dirichlet distribution setting. The competing relationships exist in the following pairs of two nodes: $(v_0, v_3), (v_1, v_9), (v_2, v_7), (v_3, v_4), (v_3, v_7), (v_4, v_6), (v_4, v_8), (v_5, v_7)$, and $(v_5, v_8)$. This is illustrated in Figure 5(a). The final collaboration relationships among FL-PTs are illustrated in Figure 5: Figure 5(b) illustrates the results for FedEgoists while Figure 5(c) illustrates the results for the baseline methods. As shown, FedEgoists can facilitate the collaboration among FL-PTs and thus achieves a better performance.

### B.4.2 The Effect of Data Heterogeneity

Data heterogeneity is typically simulated by a pathological or Dirichlet distribution. For example, the pathological distribution is used in [5, 42]. In this paper, we conducted experiments on pathological and Dirichlet distributions respectively. In addition, We also verify the effect of the data heterogeneity level on the effectiveness of the proposed solution. Specifically, let $m$ denote the number of classes. In Dirichlet distribution, there is a distribution vector $q_c \in \mathbb{R}^m$ is drawn from the Dirichlet distribution $Dir_m(\beta)$ for each class $c$ and FL-PT $v_i$ is allocated a $q_{c,i}$ proportion of data samples of class $c$; smaller $\beta$ value results in higher data heterogeneity. We vary the value of $\beta$ that takes different values in $\{0.01, 0.1, 0.5\}$ and conducted the corresponding experiments. The related experimental results are presented in Table 5. It is observed that, in some cases of high data heterogeneity, FedEgoists achieves a performance close to the Local approach; overall, FedEgoists performs the best when it is compared to the baseline approaches.

## B.5 Real-world Collaboration Example

Figure 6(a) illustrates the generated benefit graph $\mathcal{G}_b$. Figure 6(b) illustrates the final set $\pi$ of coalitions returned by Algorithm 1 while Figure 6(c) illustrates the final set $\pi$ of coalitions of the baseline methods. Still, it is observed that, FedEgoists can facilitate the collaboration among FL-PTs and thus achieves a better performance.

## B.6 Synthetic Data

We present our experimental results with synthetic data across different scenarios with fixed competing graphs like [42]. There are $n = 8$ FL-PTs. The synthetic data are generated by $x \sim \mathcal{U}[-1.0, 1.0]$. The noise $\epsilon \sim \mathcal{N}[0.0, 0.5^2]$ is added to each label. Given the FL-PT $v_i$, the grand truth weights $u_{i,l} = v + r_{i,l}$ are sampled as $v \sim \mathcal{U}[0.0, 1.0]$ and $r_{i,l} \sim \mathcal{N}[0.0, \rho^2]$ where $l = 1, 2, 3$; $\rho^2$ measures the data distribution discrepancy among FL-PTs. We set $\rho = 0.01$, which means that the generated data are weakly non-iid in terms of sample features and labels. We construct the hypernetwork using a 1-layer MLP for training the Pareto Front of all objectives. The target network consists of a two-layer linear network. After the first linear layer, a Leaky ReLU activation function is applied.

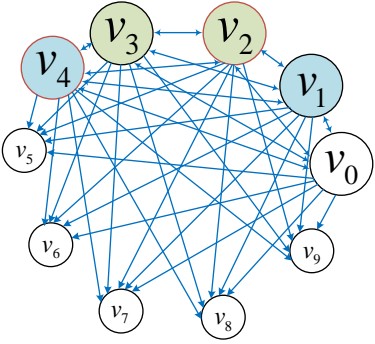

(a) The Benefit Graph $\mathcal{G}_b$.

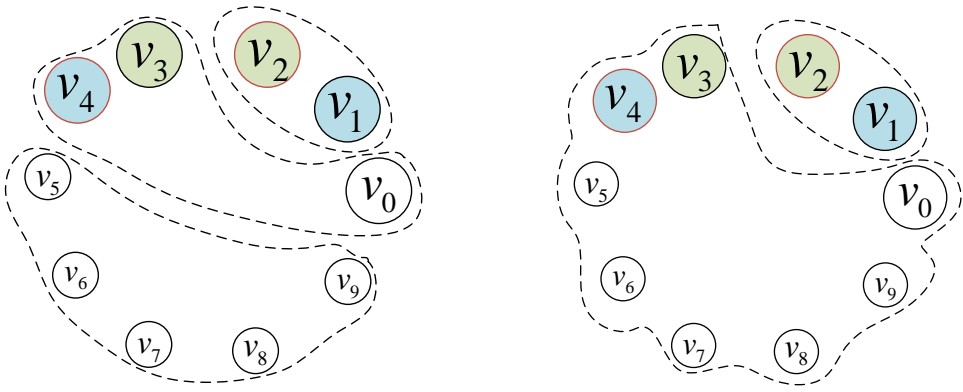

(b) The set $\pi$ of coalitions in the baseline algorithms.

(c) The set $\pi$ of coalitions by Algorithm 1.

Figure 6: **Real-world Collaboration Example**

**Weakly Non-IID setting.** The same regression task is learned by all FL-PTs and the synthetic labels are defined as:

$$y = \sum\nolimits_{k=1}^{3} u_{i,k}^T x^k + \epsilon. \tag{11}$$

FL-PTs $v_1$, $v_2$, $v_5$ and $v_6$ have 2000 samples whose amount is large, while the other FL-PTs have 100 samples whose amount is small. Thus, there exists quantity skew, i.e., a significant difference in the sample quantities of FL-PTs. Two large FL-PTs $v_1$ and $v_2$ are independent and competes with the other two large FL-PTs $v_5$ and $v_6$ that are independent. Each small FL-PT competes one large FL-PT: $(v_1, v_7)$, $(v_2, v_8)$, $(v_3, v_5)$, and $(v_4, v_6)$ are edges in the competing graph $\mathcal{G}_c$. Such $\mathcal{G}_c$ leads to a unique clique cover. Under this setting, small FL-PTs benefit large FL-PTs little. The experiments results (measured by mean squared error (MSE)) are given in Table 6.

**Strongly Non-IID setting.** We generate conflicting learning tasks by flipping over the labels of some FL-PTs:

$$y = \sum\nolimits_{k=1}^{3} u_{i,k}^T x^k + \epsilon, i \in \{1,2,3,4\} \tag{12}$$

$$y = -\sum\nolimits_{k=1}^{3} u_{i,k}^T x^k + \epsilon, i \in \{5,6,7,8\} \tag{13}$$

Different from the Weakly Non-IID setting, each FL-PT has 2000 samples, implying that there is no quantity skew. The setting of flipped labels between Eq. (12) and (13) leads to strongly Non-IID among the eight FL-PTs. We test on a different competing graph where there are two independent groups of FL-PTs $\{v_i\}_{i=1}^4$ and $\{v_i\}_{i=5}^8$: for $i \in \{1,5\}$, the FL-PTs $v_i$ and $v_{i+1}$ are independent and compete with $v_{i+2}$ and $v_{i+3}$ that are independent. Under this setting, all FL-PTs can benefit each other. The experiments results are given in Table 7.

Table 6: Experimental results (MSE) with synthetic data under fixed competing graphs: The weakly non-IID setting

| | $v_1$ | $v_2$ | $v_3$ | $v_4$ | $v_5$ | $v_6$ | $v_7$ | $v_8$ |
|---|---|---|---|---|---|---|---|---|
| LOCAL | $0.32 \pm 0.05$ | $0.28 \pm 0.00$ | $1.00 \pm 0.07$ | $0.69 \pm 0.08$ | $0.28 \pm 0.02$ | $0.28 \pm 0.01$ | $0.72 \pm 0.06$ | $0.90 \pm 0.11$ |
| FEDAVE | $0.25 \pm 0.01$ | $0.25 \pm 0.01$ | $0.79 \pm 0.05$ | $0.55 \pm 0.05$ | $0.23 \pm 0.01$ | $\mathbf{0.23 \pm 0.00}$ | $0.61 \pm 0.04$ | $0.74 \pm 0.07$ |
| FEDPROX | $0.26 \pm 0.01$ | $0.27 \pm 0.01$ | $0.90 \pm 0.10$ | $0.67 \pm 0.06$ | $0.26 \pm 0.01$ | $0.26 \pm 0.01$ | $0.76 \pm 0.11$ | $1.02 \pm 0.17$ |
| SCAFFOLD | $0.27 \pm 0.01$ | $0.28 \pm 0.00$ | $0.90 \pm 0.03$ | $0.67 \pm 0.06$ | $0.25 \pm 0.01$ | $0.26 \pm 0.01$ | $0.72 \pm 0.09$ | $0.92 \pm 0.10$ |
| PFEDME | $0.28 \pm 0.02$ | $0.29 \pm 0.03$ | $1.13 \pm 0.55$ | $0.86 \pm 0.58$ | $0.33 \pm 0.13$ | $0.33 \pm 0.12$ | $0.74 \pm 0.02$ | $0.82 \pm 0.04$ |
| PFEDHN | $0.35 \pm 0.07$ | $0.31 \pm 0.05$ | $0.91 \pm 0.07$ | $0.61 \pm 0.06$ | $0.33 \pm 0.04$ | $0.31 \pm 0.05$ | $0.70 \pm 0.09$ | $0.90 \pm 0.18$ |
| PFEDGRAPH | $0.26 \pm 0.01$ | $0.27 \pm 0.01$ | $0.90 \pm 0.04$ | $0.67 \pm 0.08$ | $0.26 \pm 0.01$ | $0.26 \pm 0.00$ | $0.74 \pm 0.08$ | $0.99 \pm 0.05$ |
| FEDEGOISTS | $\mathbf{0.23 \pm 0.01}$ | $\mathbf{0.24 \pm 0.00}$ | $\mathbf{0.24 \pm 0.01}$ | $\mathbf{0.22 \pm 0.02}$ | $\mathbf{0.22 \pm 0.00}$ | $0.23 \pm 0.01$ | $\mathbf{0.25 \pm 0.01}$ | $\mathbf{0.25 \pm 0.02}$ |

Table 7: Experimental results (MSE) with synthetic data under fixed competing graphs: The strongly non-IID setting

| | $v_1$ | $v_2$ | $v_3$ | $v_4$ | $v_5$ | $v_6$ | $v_7$ | $v_8$ |
|---|---|---|---|---|---|---|---|---|
| LOCAL | $0.29 \pm 0.03$ | $0.29 \pm 0.02$ | $0.26 \pm 0.00$ | $0.29 \pm 0.04$ | $0.27 \pm 0.01$ | $0.27 \pm 0.04$ | $0.27 \pm 0.02$ | $0.27 \pm 0.01$ |
| FEDAVE | $0.25 \pm 0.00$ | $\mathbf{0.25 \pm 0.01}$ | $\mathbf{0.23 \pm 0.01}$ | $0.23 \pm 0.01$ | $0.23 \pm 0.01$ | $\mathbf{0.22 \pm 0.00}$ | $0.23 \pm 0.02$ | $0.24 \pm 0.02$ |
| FEDPROX | $0.27 \pm 0.02$ | $0.26 \pm 0.01$ | $0.26 \pm 0.01$ | $0.26 \pm 0.01$ | $0.24 \pm 0.01$ | $0.24 \pm 0.01$ | $0.25 \pm 0.01$ | $0.25 \pm 0.01$ |
| SCAFFOLD | $0.26 \pm 0.01$ | $0.26 \pm 0.01$ | $0.26 \pm 0.01$ | $0.26 \pm 0.01$ | $0.24 \pm 0.01$ | $0.24 \pm 0.01$ | $0.25 \pm 0.01$ | $0.25 \pm 0.01$ |
| PFEDME | $0.36 \pm 0.12$ | $0.37 \pm 0.12$ | $0.25 \pm 0.00$ | $0.25 \pm 0.01$ | $0.28 \pm 0.02$ | $0.27 \pm 0.01$ | $0.27 \pm 0.01$ | $0.28 \pm 0.01$ |
| PFEDHN | $0.33 \pm 0.05$ | $0.34 \pm 0.03$ | $0.32 \pm 0.05$ | $0.28 \pm 0.03$ | $0.34 \pm 0.03$ | $0.29 \pm 0.03$ | $0.29 \pm 0.05$ | $0.29 \pm 0.06$ |
| PFEDGRAPH | $0.26 \pm 0.01$ | $0.27 \pm 0.01$ | $0.26 \pm 0.02$ | $0.26 \pm 0.02$ | $0.24 \pm 0.01$ | $0.24 \pm 0.01$ | $0.25 \pm 0.01$ | $0.25 \pm 0.01$ |
| FEDEGOISTS | $\mathbf{0.24 \pm 0.00}$ | $0.27 \pm 0.05$ | $0.24 \pm 0.03$ | $\mathbf{0.22 \pm 0.01}$ | $\mathbf{0.22 \pm 0.00}$ | $\mathbf{0.22 \pm 0.00}$ | $\mathbf{0.22 \pm 0.01}$ | $\mathbf{0.22 \pm 0.01}$ |

## B.7 Computer Resources

The system is equipped with an Intel(R) Xeon(R) Gold 6148 CPU operating at 2.40GHz. It utilizes 16 Nvidia Tesla V100 GPUs, each with 32GB of memory. The installed CUDA version is 11.7, and the graphics driver version is 515.48.07.

