# OpenReview forum: "Free-Rider and Conflict Aware Collaboration Formation for Cross-Silo Federated Learning"
_NeurIPS.cc/2024/Conference — NeurIPS 2024 poster_

### Official Review · Reviewer_2xqy · 2024-06-15

**Soundness:** 2
**Presentation:** 2
**Contribution:** 2
**Rating:** 6
**Confidence:** 4

**Summary:**

This paper introduces a strategy named FedEgoists designed to enhance collaboration in cross-silo federated learning (FL) scenarios, particularly in business sectors where participants (FL-PTs) are often competitive and self-interested. The proposed FedEgoists strategy presents a sophisticated and theoretically sound approach to managing collaborations in cross-silo federated learning, and its strengths lie in its ability to handle self-interest and competition effectively.

**Strengths:**

FedEgoists ensures that the formed coalitions are optimal. No coalition can improve its utility by merging with another coalition, making the solution stable and efficient. The strategy effectively addresses the problem of free riders, ensuring that all FL-PTs contribute to and benefit from the FL ecosystem proportionately. By preventing FL-PTs from contributing to their competitors or their supporters, the strategy minimizes conflicts of interest. This paper validates FedEgoists using the benchmark experimental datasets.

**Weaknesses:**

The paper lacks rigorness as pointed out in the Questions part.

**Questions:**

(1) FedEgoists requires a central server to coordinate and enforce the coalition formation, which introduces a single point of control and potential failure. This centralization might conflict with the decentralized nature of federated learning and violate the fundamental motivation of FL.
(2) FedEgoists assumes that FL-PTs will accurately report their competitive relationships to the central server (CS). However, this can easily lead to privacy leaks and potential attacks.
(3) While the FedEgoists strategy focuses on collaboration benefits and competition avoidance, it does not explicitly address data complementarity.
(4) Since the FedEgoists strategy relies on coalition formation based on reported benefits and competition, there is a risk of strategic manipulation by participants.
(5) The experiments are not sufficient, and there are too few datasets and baselines.

**Limitations:**

It seems limitations are not clearly or explictly elaborated.

---

> ### Author Rebuttal · Authors · 2024-08-07
>
> **Comments 1.** FedEgoists requires a central server to coordinate and enforce the coalition formation, which introduces a single point of control and potential failure. This centralization might conflict with the decentralized nature of federated learning and violate the fundamental motivation of FL. Since the FedEgoists strategy relies on coalition formation based on reported benefits and competition, there is a risk of strategic manipulation by participants.
>
>
> $\color{blue}{Response.}$ In the classic federated learning (FL) framework, there is a central server that is assumed to be trustable. The decentralized nature of FL refers to the data of different FL-PTs (i.e., FL participants) being decentralized since the local data of FL-PTs don’t need to be transferred to the central server.
>
> As introduced in Section 1, in the vanilla FedAvg framework, multiple FL-PTs train a shared model locally with their own dataset, and upload their local model updates to a central server trusted by all FL-PTs, which then aggregates these model updates and distributes the model updates to each FL-PT (i.e., client) [22]. Following such a client-server architecture of FedAvg, many important techniques have been proposed to improve the FL performance while facing the statistical heterogeneity of data over FL-PTs (e.g., FedProx [18], SCAFFOLD [14], pFedHN [26], pFedMe [28], FedDisco [40], pFedGraph [39]). Such client-server architectures are especially applicable to cross-silo FL considered in this paper; here, FL-PTs are typically companies or organizations with reliable computational resources and communication channels [9,10]. In such architectures, the central server has authority to determine the contribution relationships of FL-PTs (i.e., the ways of aggregating the model updates for FL-PTs) in the FL training process (e.g., [5,6,30]).
>
> There is a development process for a technique to be implemented in real worlds and adopted by organizations. Business sectors are an important FL application domain; addressing better the concerns of FL participants in business sectors can facilitate the participation of potential organizations into a FL ecosystem. The theoretical framework of this paper sticks to the current FL technical practices and serves as a starting point to construct a FL ecosystem where FL-PTs have both self-interest and competition features and a FL manager needs to construct collaboration relationships of FL-PTs without free-riding and conflicts of interests between FL-PTs.
>
> The reviewer also raised an excellent point about a single point of control and potential failure. In some important application domains of FL such as wireless network systems, it is necessary to consider such issues with a single point of failure; as shown by the survey below, in such domains, blockchain-based federated learning can be promising where there is no central server and peer-to-peer coordination between FL-PTs is needed:
>
> - Nguyen, Dinh C., Ming Ding, Quoc-Viet Pham, Pubudu N. Pathirana, Long Bao Le, Aruna Seneviratne, Jun Li, Dusit Niyato, and H. Vincent Poor. "Federated learning meets blockchain in edge computing: Opportunities and challenges." *IEEE Internet of Things Journal* 8, no. 16 (2021): 12806-12825.
>
> In blockchain-based FL without a central server, it is an interesting research problem to design proper negotiation protocols to guarantee that there are no free-riding and conflicts of interests between FL-PTs. This will be highlighted as a nice future work in the final version of our manuscript.
>
> **Comments 2.** FedEgoists assumes that FL-PTs will accurately report their competitive relationships to the central server (CS). However, this can easily lead to privacy leaks and potential attacks.
>
>
> **$\color{blue}{Response.}$** Following the standard technical practices in literature, the central server is assumed to be trustable, as explained above. In real worlds, the central server may represent an impartial and authoritative third-party (e.g., the industry association) [5]. Then, FL-PTs can report their competitive relationships to the third-party in person and confidentiality agreements can be signed between the third-party and FL-PTs. We will clarify this in the final version of the manuscript.
>
> **Comments 3.** While the FedEgoists strategy focuses on collaboration benefits and competition avoidance, it does not explicitly address data complementarity.
>
> **$\color{blue}{Response.}$** Yes, this paper doesn’t address the technical issues related to data complementarity. It uses the existing techniques [5,30] to evaluate the data complementarity between FL-PTs, which is represented by a benefit graph. The competitive relationships between FL-PTs form a competing graph. According to the benefit graph and the competing graph, this paper aims to construct the contribution relationships between FL-PTs to guarantee no free-riding and conflicts of interests between FL-PTs, which is desired in business sectors.
>
> **Comment 4.** The experiments are not sufficient, and there are too few datasets and baselines.
>
> **$\color{blue}{Response.}$**  Due to the space limitations, please refer to global rebuttal.

---

> > ### Comment · Reviewer_2xqy · 2024-08-08
> >
> > Not fully clearing my concern but I think I can increase my rating

---

> > > ### Author Response · Authors · 2024-08-11
> > > **Thank you!**
> > >
> > > Dear Reviewer 2xqy, we thank you sincerely for your time to check our rebuttal and your positive feedbacks.

---

### Official Review · Reviewer_gR3Y · 2024-07-05

**Soundness:** 3
**Presentation:** 3
**Contribution:** 2
**Rating:** 6
**Confidence:** 4

**Summary:**

This paper studies an interesting topic, which is about cooperation and competition in federated learning. It can be used to describe or simulate the real-world scenarios. The authors propose to use graph-related techniques to formulate the relation between the local clients. They test the algorithm on the benchmark datasets and the medical dataset.

**Strengths:**

1, The research question in federated learning may have potential practical impact in the real-world setting. The cross-silo setting fits what the authors explore. Considering the cooperation and competition from the graph perspective in FL is an interesting point.

2, Statements are supported with experiment results and theory.

3, The presentation is helpful for readers including the algorithm description, diagram, and figures.

**Weaknesses:**

1, The novelty of each module, especially the graph part, may need to be clarified clearly. Also, the two principles are commonly used, if the reviewer understand correctly, an optimal formation is proposed based on these two, is that correct?

2, Most designed part is at the server side. The reviewer may not be able to observe the strong connection with federated learning. The setting itself may tell that each client is a participant in this framework. However, the introduction (motivation) and the method could be a little bit fragmented.

3, Ablation study and hyperparameter study may help verify the algorithm more deeply and soundly.

4, A little bit concerned about the baseline selection. Some contribution-related work in FL like [1], if comparable, should be considered or at least discussed.

[1] Contribution-Aware Federated Learning for Smart Healthcare

**Questions:**

1, Could you please address the concerns in the weakness part?

2, Could you clarify how the local model information is leveraged and updated in the graph part more clearly?

3, How to validate or verify the meaning of the graph construction part? Do we need to have the groundtruth of the relation of all the clients?

**Limitations:**

1, The connection of the proposed techniques should be closely associated with the setting of FL, which focuses on the iterative updates between the server and the clients.

2, The original novelty of contribution should be emphasized a little bit more.

3, More comprehensive experiments may help demonstrate the effectiveness and understand the generalization of the algorithm better.

---

> ### Author Rebuttal · Authors · 2024-08-07
>
> **Weaknesses 1 \& Limitations 2**
>
> Overall, the original novelty of contribution includes multiple aspects: (1) identifying an interesting question to study, (2) proposing a desirable solution concept for this new problem, and (3) proposing an optimal solution.
>
> Specifically, business sectors are a key FL application domain. Firstly, the concerns of FL participants are identified (Section 1). Although the principles used to address these concerns are intuitive or used in FL literature, they are simultaneously considered for the first time. Secondly, a solution concept (or a proper problem formulation) is proposed (Section 3.3) such that the resulting solution can well satisfy the needs of FL participants and help FL participants achieve the best possible ML model performances. Thirdly, an algorithm is proposed to give an optimal solution to the defined problem (Section 4).
> The aim of developing FL techniques is to make FL able to be implemented in real worlds and adopted by organizations. The proposed theoretical framework of this paper serves as a starting point to construct a FL ecosystem where FL-PTs have both self-interest and competition features and a FL manager needs to construct collaboration relationships of FL-PTs without free-riding and conflicts of interests between FL-PTs.
>
> Yes, the solution concept of this paper is proposed based on these two principles. Of the greatest relevance to this paper is Ref. [30] that only considers Principle 1 in this paper and simply gives a heuristic solution to realize Principle 1. In this paper, we are motivated by the scenario under study and consider simultaneously realizing Principle 1 as well as Principle 2. Also, a refined solution concept is proposed in Section 3.3 to guarantee that no coalitions can collaborate together and be merged into a larger one to achieve a higher utility.
>
> **Weaknesses 2 \& Limitations 1**
>
> This paper is about the application of federated learning (FL) to business sectors and addresses the related issues about free-riders and conflicts of interests, which are the key to enabling the application of FL in the scenario under study. It is interdisciplinary in nature and may relate to the topics “Applications” and “Social and economic aspects of machine learning” in the Call for Papers of NeurIPS-2024. Its focus is not on the FL’s technical details or how to improve the potential technical flaws in the current FL study; in contrast, its focus is on the social and economic effect of FL. Applications are the destinations of a technique; business sectors are an important domain to apply FL; addressing better the concerns of FL participants in business sectors can facilitate the application of FL, thus amplifying its social and economic effect. On the other hand, in order to validate the effectiveness of the proposed theoretical framework, related FL techniques have to be implemented in the experimental evaluation part (i.e., Section 5).
>
> **Weaknesses 3**
>
> Motivated by your suggestions, more experiments have been conducted to verify the algorithm better.
>
> The experiments aim to verify the effectiveness of the proposed theoretical framework in the scenario under study. The scenario features self-interest for every client and competition between a part of clients. The collaboration relationships between clients are built according to the competing graph and the benefit graph; the latter graph depends on the data heterogeneity/complementarity of clients. Thus, the variable factors in the scenario/experimental environment include the intensity of competition between clients and the non-IID setting. Following your suggestions, we vary the related parameters in these two factors to verify the algorithm better.
>
> Firstly, two clients are either independent of each other, or compete against each other. For each pair of clients, there is a probability $\alpha$ that these two clients compete; here the probability that these two clients are independent of each other is $1 - \alpha$. The value of $\alpha$ determines the intensity of competition among clients and a larger value reflects a higher level of competing intensity among clients. In the benchmark experiments on CIFAR-10 and CIFAR-100, we better verify the effect of the competition intensity on the effectiveness of the proposed solution where we vary the value of $\alpha$ that takes different values in {0.05,0.1,0.2,0.3,0.4} and conducted the corresponding experiments. The related experimental results are presented in Tables 1 & 2. All the tables can be found in the PDF document of the global rebuttal.
>
> Secondly, the data heterogeneity is typically simulated by a pathological or Dirichlet distribution. e.g., the pathological distribution is used in [5,30]. In this paper, we conducted experiments on pathological and Dirichlet distributions, respectively. Also, in this rebuttal, we better verify the effect of data heterogeneity on the effectiveness of the proposed solution. Specifically, let $m$ denote the number of classes. In Dirichlet distribution, there is a distribution vector $q_{c}\in \mathbb{R}^{m}$ is drawn from the Dirichlet distribution $Dir_{m}(\beta)$ for each class $c$ and client $v_{i}$ is allocated a $q_{c,i}$ proportion of data samples of class $c$; smaller $\beta$ value results in higher data heterogeneity. We vary the value of $\beta$ that takes different values in {0.01,0.1,0.5} like [39] and conducted the corresponding experiments. The related experimental results are presented in Table 4.
>
> Besides the added experiments, we better clarify the type or the research focus of this paper. In the experiments, since this paper doesn’t focus on researching the FL technique itself, we simply follow the current best technical practices in FL literature. Like [5,30], ablation study and hyperparameter study are not taken and this paper doesn’t focus on the design of network architectures.

---

> ### Author Response · Authors · 2024-08-07
>
> **Weaknesses 4**
>
> The two papers have different focuses. There is a central server that coordinates the FL training process. In [1], a global ML model is built for all clients and the paper aims to evaluate the individual contribution of each client/FL-PT to this global model. In our paper, a personalized ML model is produced for each individual client $v_{i}$ and the central server determines which clients can contribute to this client $v_{i}$ in the FL training process. While determining such contribution relationships between clients, we aim to avoid free-riding and conflicts of interests between clients.

---

> ### Author Response · Authors · 2024-08-07
>
> **Questions 2 \& 3**
>
> This paper has two graphs (i.e., the benefit graph $\mathcal{G_b}$, and the competing graph $\mathcal{G_c}$) that are assumed to be known to a FL manager; $\mathcal{G_b}$ is obtained by collaborative model training, while the information on $\mathcal{G_c}$ is reported by clients to the FL manager (or the central server). Below, we detail the ways of obtaining them and how the local model information is leveraged in the graph part.
>
> Firstly, data heterogeneity/complementarity between clients is characterized by a benefit graph $\mathcal{G_b}$. Like [5,30], this paper uses the hypernetwork technique in [23] to evaluate the data complementarity, thus obtaining the benefit graph. In the graph part, the local model information is mainly leveraged in the calculation of $\mathcal{G_b}$. We also refer readers to Section 4.2 of Ref. [5] for the technical details on the way of obtaining $\mathcal{G_b}$.
>
> Specifically, there are $n$ clients. Each client $v_{i}$ has a risk/loss function $\ell_{i}$: $\mathbb{R^n}\rightarrow \mathbb{R_+}$. Given a learned hypothesis $h\in$ $\mathcal{H}$, let the loss vector $\mathbf{\ell}(h)=[\ell_{1}, \dots, \ell_{n}]$ represent the utility loss of the $n$ clients under the hypothesis $h$. The hypothesis $h$ is considered a Pareto solution if there is no other hypothesis $h^{\prime}$ that dominates $h$, i.e., $\nexists h^{\prime}\in \mathcal{H}, s.\,t.\, \forall i: \ell_{i}(h^{\prime})\leqslant \ell_{i}(h) \text{ and } \exists j: \ell_{j}(h^{\prime}) < \ell_{j}(h). $
> Let $r=(r_{1}, \dots,r_{n})$ $\in \mathbb{R^n}$ denote a preference vector which denotes the weight of the objective local model loss that is normalized with $\sum_{k=1}^{n}{r_{k}}=1$ and $r_{k}\geq 0, \forall k\in \{1, \dots, n\}$. The hypernetwork $HN$ takes $r$ as input and outputs a Pareto solution $h$, i.e., $h\gets HN(\phi, r)$, where $\phi$ denotes the parameters of the hypernetwork [23]. For each client $v_{i}$, linear scalarization can be used. Like [5,30], an optimal preference vector $r_{i}^{\ast}=\left(r_{i,1}^{\ast}, r_{i,2}^{\ast}, \dots, r_{i,n}^{\ast}\right)$ is determined to generate the hypothesis $h_{i}^{\ast}$ that minimizes the loss with the data $\hat{\mathcal{D_i}}$. This is expressed as
> $h_{i}^{\ast}=HN(\phi, r_{i}^{\ast}), \text{ where } r_i^* = argmin_r \hat{\mathcal{L_i}}(HN(\phi, r)).$
> For each client $v_{i}$, the value of $r_{i,j}^{\ast}$ is used as an estimate to the weight of $v_{j}$ to $v_{i}$ [5,30]. $r_{1}^{\ast}, r_{2}^{\ast}, \dots, r_{n}^{\ast}$ define a directed weighted graph, i.e., the benefit graph $\mathcal{G_b}$.
>
> Secondly, following the standard technical practices in literature, the central server is assumed to be trustable, as explained above. In real worlds, the central server may represent an impartial and authoritative third-party (e.g., the industry association) [5]. Then, FL-PTs can report their competitive relationships to the third-party in person and confidentiality agreements can be signed between the third-party and FL-PTs.
>
> The above information will be clarified in the final version of our manuscript.

---

> ### Author Response · Authors · 2024-08-07
>
> **Limitations 3**
>
> Due to space limitations, please refer to the global rebuttal.

---

> > ### Comment · Reviewer_gR3Y · 2024-08-08
> >
> > Thanks for the author's comments. I think the comments have addressed my concerns and answered the questions. I will increase the score.

---

> > > ### Author Response · Authors · 2024-08-11
> > > **Thank you!**
> > >
> > > Dear Reviewer gR3Y, we thank you sincerely for your time to check our rebuttal. We are glad that our response has addressed your concerns.

---

### Official Review · Reviewer_BcXD · 2024-07-07

**Soundness:** 3
**Presentation:** 4
**Contribution:** 3
**Rating:** 7
**Confidence:** 5

**Summary:**

The business sector is a main domain where cross-silo federated learning (FL) has many promising applications in various scenarios. The authors simultaneously consider the self-interest and competition features in the business sector. They develop a novel framework to both address the resulting free-riding problem and avoid the conflict of interest between any two competing FL participants (FL-PTs), which results in an interesting optimization problem. Finally, the authors find an optimal solution where one coalition cannot increase the utility of any of its members by collaborating with other coalitions. Extensive experiments are conducted to show the effectiveness of the proposed solution, and its ability to establish efficient collaborative networks in cross-silo FL with FL-PTs that engage in business activities.

**Strengths:**

S1: The authors consider a timely and interesting question in cross-silo FL where the free-riding and competition issues are considered simultaneously.
S2: The authors propose a new problem formulation and definition and develop a novel theoretical framework of practical importance to address the identified problem. The resulting optimization problem is solved optimally. The paper is technically sound.
S3: The paper is well organized and easy to follow. The background and related work are well introduced to understand the importance of the problem of this paper.
S4: Extensive experiments have been done to show the effectiveness of the proposed solution.

**Weaknesses:**

W1: The proposed algorithm is not polynomial-time solvable. However, this seems reasonable since there are typically a limited number of FL-PTs (e.g., 2 to 100) for cross-silo FL in business sectors. The authors propose a novel application of the classic algorithms in graph theory to solve their problem optimally. The algorithm complexity depends on these classic graph algorithms, which work well in reality.

**Questions:**

Q1: Can the proposed technical framework be applied to other types of tasks or datasets, e.g., NLP? In the experimental evaluation part, the authors have already shown its application to some interesting tasks commonly seen in literature.
Q2: To keep consistency, it is better that “selfish” in Figure 1 is changed to “self-interest”.
Q3: In this paper, all FL-PTs are partitioned into mutually disjoint coalitions/groups. It seems that this is similar to clustered federated learning where FL-PTs are partitioned into mutually disjoint clusters (i.e., groups). The reviewer understands that the authors consider a problem that is clearly different from the typical clustered federated learning.
Q4: In your experimental results, how many trials have you conducted to obtain the standard deviation?

**Limitations:**

See weaknesses and Questions.

---

> ### Author Rebuttal · Authors · 2024-08-07
>
> The authors would like to thank you sincerely for your overall positive comments on the manuscript, including your positive acknowledgement of the question under study, the theoretical soundness of the proposed framework, and the effective experimental validation of the proposed solution.
>
> **Q1:** The method itself is general and is expected to be applicable to NLP field, which will be shown in a future journal version.
>
> **Q2:** In the final version, we will update the manuscript as you suggested.
>
> **Q3:** In the final version, we will better clarify in Section 2 the connection and differences of our work with clustered federated learning.
>
> **Q4:** In the experiments, five trials are conducted to obtain the standard deviation. We will clarify this in the final version.

---

> > ### Comment · Reviewer_BcXD · 2024-08-12
> >
> > Thanks for the detailed responses. I'm generally fine with the results.  Thus, I will keep my score.

---

> > > ### Author Response · Authors · 2024-08-12
> > > **Thank you!**
> > >
> > > Dear Reviewer BcXD, we thank you sincerely for your time to check the rebuttal and your recognization of our work.

---

### Official Review · Reviewer_jdA3 · 2024-07-13

**Soundness:** 3
**Presentation:** 3
**Contribution:** 3
**Rating:** 4
**Confidence:** 4

**Summary:**

This paper focuses on client selection in cross-silo federated learning. The authors propose FedEgoists. In particular, FedEgoist participate clients into different clusters to avoid free riders and conflict of interests. Theoretical analysis is provided to validate the theoretical soundness of the proposed FedEgoist.
Experiments validate the effectiveness of proposed FedEgoist.

**Strengths:**

* The discussed topic, client selection in cross-silo federated learning, is important in practical setting.
* The writing is easy to follow.

**Weaknesses:**

* The evaluation setting on CIFAR-10 and CIFAR-100 cannot simulate conflict of interests in the real world. Further consideration needed.
* The meaning of propositions 1 & 2 is unclear. Further explanation is expected.

**Questions:**

Please refer to weaknesses.

**Limitations:**

There is no potential negative societal impact of their work.

---

> ### Author Rebuttal · Authors · 2024-08-07
>
> The authors would like to thank you sincerely for your overall positive comments, including the acknowledgement of the importance of the discussed topic in practical setting and the theoretical soundness of the proposed framework. Your comments are also constructive to help improve the manuscript. Below, we made a response to the two questions in the weakness part, indexed as W1 & W2 respectively.
>
>
> **W1: Experiments on CIFAR-10 & CIFAR-100**
>
> In this rebuttal, we better clarify the setting in our experiments. The setting here is the same as the setting in [30] where the same competitive relationships between clients are considered in the FL context. Two clients (i.e., FL-PTs in the paper) are either independent of each other, or compete against each other. For each pair of clients, there is a probability $\alpha$ that these two clients compete; here the probability that these two clients are independent of each other is $1 - \alpha$. The value of $\alpha$ determines the intensity of competition among clients and a larger value reflects a higher level of competing intensity among clients.
>
> Compared with [30], we define a new metric to better validate the performance of the proposed approach and conducted more experiments to show the robustness of the proposed approaches in terms of the level of competing intensity between clients.
>
> + Firstly, we show the performance of the proposed approach when $\alpha$ takes different values in {0.05,0.1,0.2,0.3,0.4}, representing different levels of competing intensity between clients. We conducted five trials to show the average performance. For the $l$-th trial, a particular competing graph $G_{c,l}$ is randomly generated for the $l$-th time with the given $\alpha$; then, the experiments for the baseline and proposed approaches are run; the performance of the proposed approach is denoted as $r_{\alpha, l, p}$ while the performance of the $i$-th baseline approach is denoted as $r_{\alpha, l, i}$. Given the value of $\alpha$, we show the average performance of the five trials (i.e., $\sum_{l=1}^{5}{r_{\alpha, l, p}}/5$ and $\sum_{l=1}^{5}{r_{\alpha, l, i}}/5$, $i=1,\dots, 9$). The related experimental results are presented in Tables 1 and 2. All tables can be found in the uploaded PDF document of the global rebuttal. For CIFAR-10, the proposed FedEgoists achieves the best performance in all cases when compared with all the basedline approaches. For CIFAR-100, the proposed FedEgoists achieves the best performance on average when compared with all the basedline approaches.
>
> + Secondly, suppose we are given a specific value of $\alpha$. We define a new metric to show the worst-case performance of the proposed approach compared with the baseline approaches by the following ways:
>    - Firstly, we find an integer $l^{\ast}$ such that under the $l^{\ast}$-th trial, the performance of the baseline approaches is the best compared with the proposed approach, i.e. $l^{\ast} = \arg\max\limits_{l\in [1, 5]}\{ (\max\limits_{i\in [1,9]}{r_{\alpha, l, i}}) - r_{\alpha, l, p} \}$ where $\max\limits_{i\in [1,9]}{r_{\alpha, l, i}}$ is the best performance of all the baseline approaches in the $l$-th trial and $(\max\limits_{i\in [1,9]}{r_{\alpha, l, i}}) - r_{\alpha, l, p}$ is their performance improvement (or the difference) to the proposed approach, which may be negative if the proposed approach achieves a better performance.
>    - Secondly, after finding such $l^{\ast}$, we show the performance of the proposed and baseline approaches under the $l^{\ast}$-th trial. Specifically, given the value of $\alpha$, we compute the value of $(\max\limits_{i\in [1,9]}{r_{\alpha, l, i}}) - r_{\alpha, l, p}$; these values under different $\alpha$ are presented in Table 3.
>
> $\color{blue}{Conclusion.}$ Table 3 shows that in the worst case, the proposed FedEgoists has a performance very close to the best performance of all the baseline approaches. Tables 1 and 2 show that on average, the proposed FedEgoists achieves a significant performance improvement when compared with all the baseline approaches.
>
>
> **W2: The Meaning of Propositions 1 & 2**
>
> You raised an excellent point about the clarity. In the final version, we will better clarify the meaning of Propositions 1 \& 2 and better show how Algorithm 1 gives an optimal solution to the problem of this paper defined in Section 3.3 (lines 183-185 on page 5).
>
> Specifically, clients are partitioned into multiple coalitions/groups, and this paper aims to find a solution that satisfies Principles 1 \& 2 and $\Pi=\emptyset$. Principles 1 \& 2 guarantees that there are no free-riders and conflicts of interests in a FL ecosystem; the requirement of $\Pi=\emptyset$ guarantees that no coalitions can collaborate together and be merged into a larger one to achieve a higher utility. In such context, the meaning of Propositions 1 \& 2 is as follows:
>    + Proposition 1 shows that Eq. (1) holds and Principle 2 is realized in the solution given by Algorithm 1. Eq. (1) is given in Section 3.2.1 where we say that _Principle 1 is realized when Eq. (1) is satisfied_, which will be highlighted as a lemma in the final version and used in the proof of Proposition 1. Also, Proposition 1 will be updated to directly say that ``_Upon completion of Algorithm 1, Principles 1 \& 2 are realized._".
>    + Proposition 2 shows that the requirement of $\Pi=\emptyset$ in Eq. (3) is satisfied.
>
> In the final version of the manuscript, at the end of Section 4, the physical meaning of the conclusions in Propositions 1 & 2 will be clarified better as explained above.

---

> ### Author Response · Authors · 2024-08-07
>
> **Notes on other added experiments.** In this rebuttal, besides the experiments conducted for the above comments in W1, more experiments are also conducted (a) to verify the robustness of the proposed solution in terms of the level of data heterogeneity, and (b) on a new dataset and with a new baseline approach.
>
> **Experiments (a)**
>
> Specifically, this paper is about the application of the current FL techniques to the business sectors, and the experiments aim to verify the effectiveness of the proposed theoretical framework in the scenario under study. The scenario features self-interest for every client and competition between a part of clients. The collaboration relationships between clients are built according to the competing and benefit graphs; the benefit graph depends on the data heterogeneity/complementarity of clients. Thus, the variable factors in the scenario/experimental environment include the intensity of competition between clients and the non-IID setting. In the experiments, we vary the related parameters in these two factors to better verify the robustness of the proposed algorithm. Above, the experiments for the intensity of competition have been introduced.
>
> Data heterogeneity is typically simulated by a pathological or Dirichlet distribution. For example, the pathological distribution is used in [5,30]. In this paper, we conducted experiments on pathological and Dirichlet distributions, respectively. Also, in this rebuttal, we better verify the effect of data heterogeneity on the effectiveness of the proposed solution. Specifically, let $m$ denote the number of classes. In Dirichlet distribution, there is a distribution vector $q_{c}\in \mathbb{R}^{m}$ is drawn from the Dirichlet distribution $Dir_{m}(\beta)$ for each class $c$ and client $v_{i}$ is allocated a $q_{c,i}$ proportion of data samples of class $c$; smaller $\beta$ value results in higher data heterogeneity. We vary the value of $\beta$ that takes different values in {0.01,0.1,0.5} and conducted the corresponding experiments. The related experimental results are presented in Table 4.
>
> **Experiments (b)**
>
> We additionally conducted experiments on the synthetic data used in [30] where the experimental setting is also the same as the setting in [30]. The related experimental results are presented in Tables 5-6, which can be found in the PDF document. We add one more approach in the paper below, called  FEDORA, as the baseline:
>
> Jun Wu, Wenxuan Bao, Elizabeth Ainsworth, and Jingrui He. "Personalized federated learning with parameter propagation." In ACM KDD, 2023.
>
> FEDORA is only for image classification tasks; thus, we only conducted experiments on CIFAR-10 and CIFAR-100. The related experimental results are presented in Tables 1,2,4.

---

> ### Author Response · Authors · 2024-08-11
> **Discussion Period & Thank you!**
>
> Dear Reviewer jdA3, we thank you for your time and detailed comments. Since the discussion period will end soon, we will appreciate if you could check the rebuttal and let us know whether our response has addressed your concerns. Thank you sincerely.

---

### Author Rebuttal · Authors · 2024-08-07

More experiments have been conducted to verify the effectiveness of the proposed solution:
1. More experiments are conducted to verify the robustness of the proposed algorithm.
2. We define a new metric to better validate the performance of the proposed approach.
3. New datasets and baselines are considered.

**Added Experiments Part 1: Robustness**

The paper is about the application of the current FL techniques to the business sectors, and the experiments aim to verify the effectiveness of the proposed theoretical framework in the scenario under study. The scenario features self-interest for every client and competition between a part of clients. The collaboration relationships between clients are built according to the competing graph and the benefit graph; the latter graph depends on the data heterogeneity or complementarity of clients. Thus, the variable factors in the scenario or experimental environment include the intensity of competition between clients and the non-IID setting. We vary the related parameters in these two factors to verify the robustness of the proposed algorithm.

Below, we explain the related parameters in these two factors and the added experiments:

Firstly, two clients are either independent of each other or compete against each other. For each pair of clients, there is a probability $\alpha$ that these two clients compete; here the probability that these two clients are independent of each other is $1 - \alpha$. The value of $\alpha$ determines the intensity of competition among clients and a larger value reflects a higher level of competing intensity among clients. In the benchmark experiments on CIFAR-10 and CIFAR-100, we better verify the effect of the competition intensity on the effectiveness of the proposed solution where we vary the value of $\alpha$ that takes different values in {0.05,0.1,0.2,0.3,0.4} and conducted the corresponding experiments.Given a specific value of $\alpha$, we conducted five trials to show the average performance. For the $l$-th trial, a particular competing graph $G_{c,l}$ is randomly generated for the $l$-th time with the given $\alpha$; then, the experiments for the baseline and proposed approaches are run; the performance of the proposed approach is denoted as $r_{\alpha, l, p}$ while the performance of the $i$-th baseline approach is denoted as $r_{\alpha, l, i}$. Given the value of $\alpha$, we show the average performance of the five trials (i.e., $\sum_{l=1}^{5}{r_{\alpha, l, p}}/5$ and $\sum_{l=1}^{5}{r_{\alpha, l, i}}/5$, $i=1,\dots, 9$).
The related experimental results are presented in Tables 1 and 2.

Secondly, the data heterogeneity is typically simulated by a pathological or Dirichlet distribution. For example, the pathological distribution is used in [5,30]. In this paper, we conducted experiments on pathological distribution(PAT.) and Dirichlet distribution(Dir), respectively. Also, in this rebuttal, we better verify the effect of data heterogeneity on the effectiveness of the proposed solution. Specifically, let $m$ denote the number of classes. In Dirichlet distribution, there is a distribution vector $q_{c}\in \mathbb{R}^{m}$ is drawn from the Dirichlet distribution $Dir_{m}(\beta)$ for each class $c$ and client $v_{i}$ is allocated a $q_{c,i}$ proportion of data samples of class $c$; smaller $\beta$ value results in higher data heterogeneity. We vary the value of $\beta$ that takes different values in {0.01,0.1,0.5} like [39] and conduct the corresponding experiments. The related experimental results are presented in Table 4.

**Added Experiments Part 2: A New Metric**

Suppose we are given a specific value of $\alpha$. We define a new metric to show the worst-case performance of the proposed approach compared with the baseline approaches by the following ways:
1. We find an integer $l^{\ast}$ such that under the $l^{\ast}$-th trial, the performance of the baseline approaches is the best compared with the proposed approach, i.e. $l^* = \arg \max_{l \in [1,5]} \left( \max_{i \in [1,9]} r_{\alpha,l,i} - r_{\alpha,l,p} \right)
$ where $\max\limits_{i\in [1,9]}{r_{\alpha, l, i}}$ is the best performance of all the baseline approaches in the $l$-th trial and $(\max\limits_{i\in [1,9]}{r_{\alpha, l, i}}) - r_{\alpha, l, p}$ is their performance improvement (or the difference) to the proposed approach, which may be negative if the proposed approach achieves a better performance.
2. After finding such $l^{\ast}$, we show the performance of the proposed and baseline approaches under the $l^{\ast}$-th trial. Specifically, given the value of $\alpha$, we compute the value of $(\max\limits_{i\in [1,9]}{r_{\alpha, l, i}}) - r_{\alpha, l, p}$; these values under different $\alpha$ are presented in Table 3.

$\color{blue}{Conclusion.}$ Table 3 shows that in the worst case, the proposed FedEgoists has a performance very close to the best performance of all the baseline approaches. Tables 1 and 2 show that on average,  FedEgoists achieves a significant performance improvement when compared with all the baseline approaches.


**Added Experiments Part 3: A New Dataset and a New Baseline**

More experiments are conducted on a new dataset and using a new baseline. Specifically, we additionally conducted experiments on the synthetic data used in [30] where the experimental setting is also the same as the setting in [30]. The related experimental results are presented in Tables 5-6. We add one more approach in the paper below, called FEDORA, as the baseline:
- Jun Wu, Wenxuan Bao, Elizabeth Ainsworth, and Jingrui He. "Personalized federated learning with parameter propagation." In *ACM KDD*, 2023.
FEDORA is only for image classification tasks; thus, we only conducted experiments on CIFAR-10 and CIFAR-100. The related experimental results are presented in Tables 1,2,4.

$\color{blue}{Remark.}$  All experimental results (tables) can be obtained from the uploaded PDF document. The results are presented in the form of mean ± std.

---

### Decision · Program_Chairs · 2024-09-25

**Decision:**

Accept (poster)

**Comment:**

In this paper, the authors focus on the selection of federated learning participants and study the problem of free riders and conflicts of interest among competitors. Now, the paper has received four reviews: one Accept, two Weak Accept, and one Borderline Reject.

From the positive points of the reviews, the reviewers have generally agreed that (1) the paper has studied an important problem in federated learning, (2) there are some interesting results, and (3) the paper is easy to follow. However, at the same time, the reviewers have pointed out several weaknesses on the evaluation, complexity, novelty, motivation, etc.

During the rebuttal, the authors have provided feedback to each reviewer on these weaknesses, and most of the reviewers remain positive after reading the rebuttal. Moreover, the rebuttal has clarified quite a few concerns raised by the reviewers. Since the overall scores are positive and above the acceptance threshold, we recommend accepting this paper for NeurIPS 2024.